# Learning Local Search with Theoretical Indicators for Job Shop Scheduling

## Abstract

Job shop scheduling problem (JSSP), where job sequences must be assigned across multiple machines to minimize makespan under fixed routes and varying processing times, is one of the most challenging combinatorial optimization problems. To improve search efficiency, we propose LSI, Local Search with Indicators, a learning-based local search method for JSSP. LSI integrates scheduling-theoretic conditions as indicators into the action evaluation, enabling the policy to focus on swaps that guarantee makespan reduction. By incorporating theoretically proven conditions into the action evaluation, LSI prioritizes promising swaps rather than treating all moves equally, representing a principled improvement of makespan. Despite relying only on a lightweight multilayer perceptron (MLP) policy network, LSI achieves competitive or superior performance compared to strong state-of-the-art approaches on diverse JSSP benchmarks, offering faster inference and robust scalability without retraining. These results demonstrate the effectiveness of embedding problem-structured theoretical principles into learning-based combinatorial optimization.

## 1 Introduction

The Job Shop Scheduling Problem (JSSP) is one of the most challenging combinatorial optimization problems (COPs), known to be NP-hard (Garey et al. (1976)). Unlike routing problems—e.g., the Traveling Salesman Problem (TSP), Vehicle Routing Problem (VRP), and its capacitated variant (CVRP)—JSSP requires each job to follow a fixed machine route with predetermined processing times. Each job consists of a sequence of operations, each of which must be processed on a designated machine for a specified duration.

Although recent methods have used reinforcement learning (RL) or imitation learning (IL) to learn complex encoder architectures such as convolutional neural networks (CNNs) (Liu et al. (2020); Han & Yang (2020)), recurrent neural networks (RNNs) including long short-term memory (LSTM) networks (Monaci et al. (2024); Iklassov et al. (2023)) and Transformer models (Zhao et al. (2022); Chen et al. (2022)), and graph neural networks (GNNs) (Zhang et al. (2020); Park et al. (2021b); Liu & Huang (2023); Park et al. (2021a); Lee & Kim (2022; 2024)) for JSSPs, no learning-based approach has yet reported optimal solutions for instances involving more than 15 machines and 15 jobs, indicating substantial room for further improvement.

This paper is motivated by two core research questions: *(1) Can theoretical insights into the JSSP guide the learning of better policies in learning-based optimization? (2) How can such theoretical knowledge be effectively incorporated into the policy network?* To address these questions, we focus on local search frameworks and propose a novel method that integrates theoretical makespan reduction conditions into the policy design.

There are three primary approaches to solving JSSPs: exact methods, improvement methods, and constructive methods. Exact methods, such as the branch-and-bound method, guarantee optimal solutions but often require excessive computational time, making them impractical for large-scale problems (Brucker et al. (1994)). Improvement methods iteratively enhance complete solutions through various search strategies, while constructive methods sequentially build solutions by assigning operations step by step. Given the prohibitive runtime of exact methods, most practical solvers rely on heuristics, either constructive or improvement. Constructive methods have attracted significant attention in time-critical scenarios due to their rapid sequential decision-making capabilities. Recent

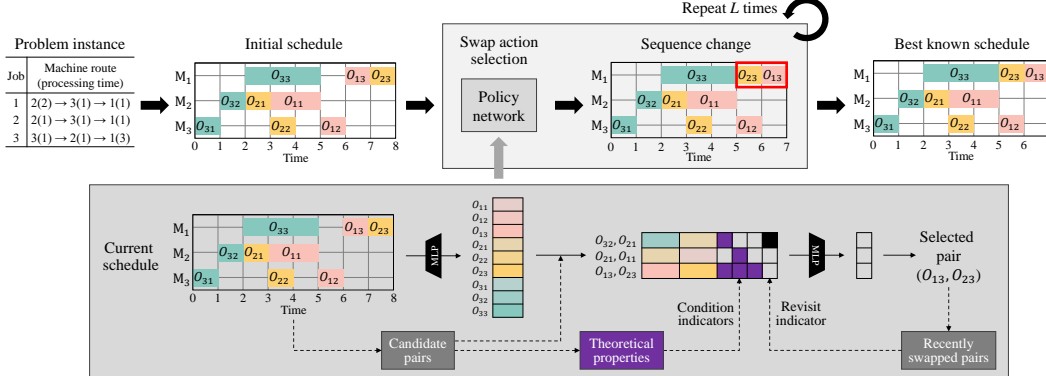

Figure 1: Overview of Local Search with Indicators (LSI), our proposed learning-based local search method for JSSPs. It begins with a given initial schedule, then iteratively swaps a pair of operations. To select a promising swap action, the policy network concatenates the embedding vectors, the theoretically derived condition indicators, and the tabu indicator of each candidate operation pair, then evaluates it. $O_{ij}$ denotes the $j$-th operation of job $i$.

studies (Zhang et al. (2020); Park et al. (2021b); Liu & Huang (2023); Park et al. (2021a); Lee & Kim (2022; 2024)) have applied GNNs combined with RL or IL to assign operations in real time. Although these methods generate feasible assignments immediately, they generally yield lower-quality solutions compared to improvement methods, which refine entire schedules through iterative adjustments. While constructive methods excel in speed, their solution quality still lags behind that of improvement methods, motivating a closer look at improvement methods. Improvement methods, such as population-based approaches including Genetic Algorithms (GA) and Particle Swarm Optimization (PSO), and local search approaches including Simulated Annealing (SA), Tabu Search (TS), and Variable Neighborhood Search (VNS), are known to produce high-quality solutions, although they tend to consume more computational time than constructive methods. Recently, learning-based local search methods have emerged, leveraging GNN-based encoders trained via RL to identify promising pairs of consecutive operations on the same machine (Falkner et al. (2022); Zhang et al. (2024a;b)).

Several studies have attempted to incorporate theoretical insights into learning-based approaches for solving JSSPs. Lee & Kim (2024) proposed a learning-based constructive method with a compact action space by ensuring reachability to an optimal schedule. Additionally, recent learning-based local search methods (Falkner et al. (2022); Zhang et al. (2024a;b)) adopt the critical path-based $N5$ neighborhood structure similar to a traditional local search approach (Nowicki & Smutnicki (1996)). This structure considers all consecutive operation swaps that could potentially reduce the makespan of the current schedule as candidate moves to generate neighbor solutions (Kuhpfahl & Bierwirth (2016)). Although the policies learn to select promising moves from the $N5$ neighbors, they treat all candidates as equally likely. They do not incorporate theoretical criteria that can distinguish which candidate moves are more likely to reduce the makespan, leaving room for further improvement.

**Novelty.** In contrast to previous learning-based local searches that rely on complex GNN architectures and treat $N5$ neighbors as a black-box, we propose a lightweight approach that incorporates theoretically grounded indicators into the policy network using only multilayer perceptrons (MLPs), as illustrated in Figure 1. We identify three novel necessary conditions for makespan reduction in JSSP and encode them as binary indicators to guide action selection. This principled integration of problem-specific knowledge enables our method to achieve superior performance on standard JSSP benchmarks, while offering faster inference and improved scalability.

## 2 PRELIMINARIES

**Job shop scheduling problem (JSSP).** JSSP is a classical NP-hard combinatorial optimization problem, proven to be NP-hard (Garey et al. (1976)). It is frequently encountered in complex manufacturing environments, such as semiconductors (Gupta & Sivakumar (2006)) or battery production

systems (Liu et al. (2021)). A JSSP instance comprises a set of $N$ jobs and $M$ machines. Each job is composed of a sequence of operations with predefined processing orders and machine assignments. Each operation must be processed on a specified machine for a given processing time. There is a precedence constraint between successive operations within each job, and machines can process only one operation at a time. Among the various objective functions considered for JSSPs, minimizing the maximum completion time, also known as makespan, is the most prevalent, as it leads to better system utilization and overall productivity (Xiong et al. (2022)). Consequently, this study aims to minimize the makespan $C_{\max}$.

**Critical path.** In a feasible schedule, a path is a directed chain of consecutive operations induced by job–route precedence and machine–processing order, from the earliest start to the latest completion. The critical path is any longest path whose length (sum of processing times) equals the makespan (Kuhpfahl & Bierwirth (2016)). For example, the sequence $O_{31}$–$O_{32}$–$O_{21}$–$O_{11}$–$O_{12}$–$O_{13}$–$O_{23}$ is the critical path of the initial schedule illustrated in Figure 1, where $O_{ij}$ denotes the $j$-th operation of job $i$. There is no idle time between consecutive operations on the critical path, and each operation on the path satisfies EST = LST, where EST (earliest start time) is the earliest possible time to start the operation in the current schedule, and LST (latest start time) is the latest possible time to start it without delaying the makespan.

**Neighborhood structures.** A key factor in the effectiveness of local search approaches is the choice of neighbors to explore. In the context of JSSPs, critical path-based neighborhood structures, denoted $N1$ (Van Laarhoven et al. (1992)), $N2$ (Dell'Amico & Trubian (1993)), and $N5$ (Nowicki & Smutnicki (1996)), are proposed to solve JSSPs and generate neighbor schedules by swapping consecutive operations on the same machine along the critical path. The inclusion relationships among $N1$, $N2$, and $N5$ are illustrated in Figure 2. The $N1$ neighborhood structure considers all such adjacent swaps and is complete,

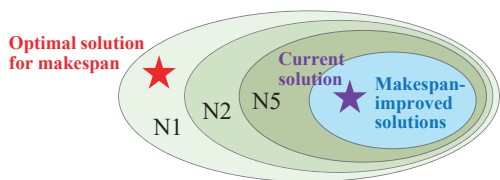

Figure 2: Schedule groups that can be moved by each neighborhood structure from the current schedule.

meaning that any optimal schedule can be reached from any arbitrary initial solution through only $N1$ moves Van Laarhoven et al. (1992). The $N2$ structure reduces the number of candidate swaps by pruning less promising ones Dell'Amico & Trubian (1993), and $N5$ prunes even further by keeping only those swaps that may reduce the makespan Nowicki & Smutnicki (1996). Empirical studies have shown that $N5$ captures most of the improvement benefits of $N1$ at a fraction of the evaluation overhead (Nowicki & Smutnicki (1996); Kuhpfahl & Bierwirth (2016)). Further explanations and illustrative examples of these neighborhood structures are provided in Appendix B.

## 3 RELATED WORKS

**Classical local search for solving JSSPs.** Local search methods have been widely used to solve JSSPs due to their ability to effectively explore the solution space (Nowicki & Smutnicki (1996); Van Laarhoven et al. (1992); Dell'Amico & Trubian (1993); Glover (1986); Mladenović & Hansen (1997); Zhang et al. (2007)). These approaches typically begin with a given solution and iteratively refine it until a stopping criterion such as a time limit or convergence threshold is met. Each local search method employs a strategy to escape local optima by occasionally accepting worse solutions. SA probabilistically accepts worse solutions, with an adaptable acceptance probability as the search progresses (Van Laarhoven et al. (1992)). TS maintains a tabu list of recent explored moves to prevent revisiting of solutions and focuses exploration on unexplored neighborhoods (Glover (1986)), and VNS perturbs the current solution if no improvement is found in a predefined number of iterations (Mladenović & Hansen (1997)). Although highly effective on small instances, their reliance on fixed neighborhood structure and manually tuned rules, such as acceptance probability, tabu list size, and perturbation method, can limit adaptability and performance in more diverse or large-scale settings.

**Learning-based local search for solving JSSPs.** Building on these classical ideas, recent learning-based methods employ RL to learn the promising selection of moves among candidates based on

critical paths using GNN-based policy networks (Falkner et al. (2022); Zhang et al. (2024a;b)). NeuroLS integrates an Implicit Quantile Network (IQN) within a VNS framework: when the makespan has not improved for two consecutive moves, the policy perturbs the schedule by randomly sampling from the larger $N1$ neighbors (Falkner et al. (2022)). Its state representation includes features such as the current and best makespans, and counts of non-improving steps and perturbations. L2S adopts a TS framework with an adaptive tabu list size, inspired by Zhang et al. (2007), and trains its GNN-based policy using $n$-step REINFORCE. Each operation is represented by features including its EST, LST, and processing time (Zhang et al. (2024a)). TBGAT further enriches this approach with a bidirectional topological Graph Attention Network (bi-GAT): one pass encodes the forward topological rank, EST, and processing time, while the other pass uses the backward topological rank and LST (Zhang et al. (2024b)). The combined embeddings enhance the model's ability to capture structural awareness. Although recent learning-based approaches have driven significant advances in JSSP performance, they share three notable limitations: (1) they treat all candidate swaps as equally probable, without leveraging theoretical distinctions in their potential to reduce makespan, (2) rely on computationally intensive GNN encoders, which slow down inference and hinder scalability to larger instances, and (3) only incorporate hand-engineered features that are not explicitly aligned with makespan reduction conditions.

To address these issues, we propose a novel framework that explicitly incorporates theoretical insights into the policy network. We derive three novel necessary conditions for makespan reduction in JSSP and prove that their joint satisfaction provides a sufficient condition in the special case of a single critical path. These theoretically grounded conditions are integrated as binary indicators into a lightweight policy network composed solely of MLPs. By avoiding complex GNN architectures, our approach significantly reduces model complexity and accelerates inference, thereby enhancing scalability. Despite the simplicity of the architecture, our method achieves state-of-the-art (SOTA) performance across JSSP benchmarks. These results highlight the effectiveness of integrating theoretical insights into the design of learning-based combinatorial optimization methods.

# 4 PROPOSED METHOD

We propose **Local Search with Indicators (LSI)**, a learning-based local search framework that integrates three theoretically derived conditions for makespan reduction as binary indicators within the policy network. These indicators help guide the selection of promising swap actions at each step of the local search. LSI operates under the standard local search framework. Starting from an initial schedule generated by a simple dispatching rule, it iteratively updates the schedule by selecting and applying a swap action between consecutive operations on the same machine, as illustrated in Figure 1. At each iteration, the $N5$ neighborhood structure is used to generate candidate swaps, each candidate is evaluated by the policy network based on the embedding vector concatenated with the embeddings of involved operations, binary indicators corresponding to the theoretical conditions, and a revisit status indicator. The action with the highest selection probability is applied to update the current schedule and is then added to the recently visiting list. This process repeats until a termination condition is met, and the best schedule found so far is returned as the final output. LSI emphasizes the use of theoretical indicators to guide decisions, improving efficiency while remaining awareness of both search history and problem structure. To train the policy network through RL, we employ the $n$-step REINFORCE. To enable this, we formulate the local search process as the MDP used by L2S (Zhang et al. (2024a)).

## 4.1 MARKOV DECISION PROCESS (MDP)

**State.** The state $s_t$ at time step $t$ represents the current schedule, including the features of the operations. For each operation $u$, we consider three features: (1) processing time $p_u$, (2) EST $est_u$, and (3) LST $lst_u$. $est_u$ denotes the earliest possible time to start operation $u$ in the current schedule, and $lst_u$ denotes the latest possible time to start it without delaying the makespan of the schedule. Note that operations with equal $est_u$ and $lst_u$ are on the critical path.

**Action.** An action $a_t$ is defined as a swap between two consecutive operations on the same machine. The candidate action set $A_t$ at time $t$ comprises all feasible swaps.

**State transition.** When action $a_t$ is selected, the corresponding swap updates the current schedule, resulting in the next state $s_{t+1}$. The EST and LST for all operations are recomputed accordingly. The episode ends either when no further candidate actions remain ($A_t = \emptyset$) or when a predefined step limit is reached.

**Reward.** The reward function $r(s_t, a_t)$ is designed to improve the best solution found so far. It is defined as: $r(s_t, a_t) = \max(C_{\max}(s_t^*) - C_{\max}(s_{t+1}), 0)$, where $s_t^*$ denotes the best solution found up to step $t$, and $s_0^* = s_0$. The cumulative reward up to step $t$ becomes: $\sum_{t'=0}^{t} r(s_{t'}, a_{t'}) = C_{\max}(s_0) - C_{\max}(s_t^*)$. By maximizing this cumulative reward, the policy network can directly optimize the performance of the solution that will be returned.

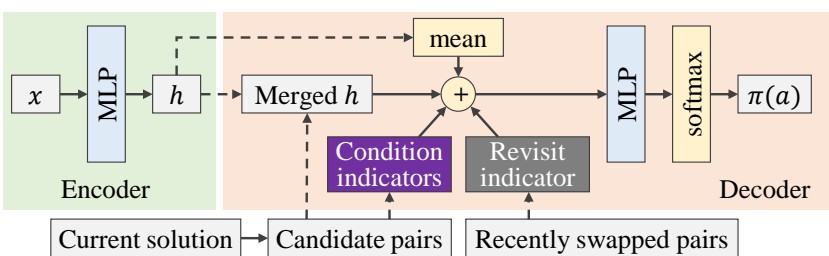

Figure 3: The architecture of the policy network.

### 4.2 ENCODER

The encoder in our model is a simple MLP. As shown in Figure 3, this MLP takes state features $x$ as input and produces a high-dimensional embedding $h_u$ for each operation $u$. We chose this architecture over complex GNN because our decoder's theoretical indicators capture the essential relational information. Ablation studies show that our simple MLP encoder enables much faster inference without losing solution quality compared to relying on complex GNN architectures.

### 4.3 DECODER

The decoder computes the action selection probabilities for each candidate action. Its key innovation is the use of three theoretically derived condition indicators whose joint satisfaction provides a sufficient condition under the restricted case of a single critical path, embedded as binary features. For every candidate swap, three binary indicators are computed, each reflecting whether the move satisfies a corresponding theoretical condition. These indicators inject problem-specific knowledge into the network, enabling more informed action selection.

Unlike L2S, LSI retains all candidates and encodes their revisit status as a binary feature in the policy network, as illustrated in Figure 1, where traditional tabu search excludes recent left moves. Also like TBGAT, the selected action $a_t$ is added to the list of recently swapped pairs, removing the oldest pair to maintain the list capacity, as described in Appendix E.

**Makespan reduction conditions in JSSP.** We present three novel propositions that describe when swapping two consecutive operations on the same machine can reduce the makespan. Unlike prior works Nowicki & Smutnicki (1996); Zhang et al. (2007); Xie et al. (2023), which define neighborhoods such as N5 that include all improving moves, shown as Figure 2, our propositions identify a smaller subset of N5 and provide a formal necessary and sufficient condition for makespan reduction when the schedule contains a single critical path.

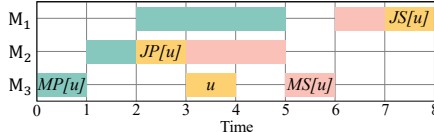

Figure 4: The directly connected operations with job or machine precedence relations for operation $u$ in a schedule, where the operations with the same color denotes the same job.

To state these precisely, we first introduce four relationships for any operation $u$. The job-predecessor of $u$, denoted $JP[u]$, is the operation immediately before $u$ in its job sequence, and the job-successor, $JS[u]$, is the operation immediately after $u$ in its job. Similarly, the machine-predecessor, $MP[u]$, is the operation scheduled just before $u$ on the same

machine, and the machine-successor, $MS[u]$, is the operation scheduled immediately after $u$ on that machine. The prime notation (e.g., $est'_u$) indicates the value after the swap, and $ect_u = est_u + p_u$ denotes the earliest completion time (ECT) of operation $u$. With these definitions in place, the proofs of the three necessary conditions are given in Appendix C.

**Proposition 1.** *When consecutive operations $u$ and $v$ are swapped on a machine of a given schedule, the makespan cannot decrease if $lst_{JS[v]} \leq ect'_v$.*

**Proposition 2.** *When consecutive operations $u$ and $v$ are swapped on a machine of a given schedule, the makespan cannot decrease if $lst_{JS[u]} \leq ect'_u$.*

**Proposition 3.** *When consecutive operations $u$ and $v$ are swapped on a machine of a given schedule, the makespan cannot decrease if $lst_{MS[v]} \leq ect'_u$.*

In the case of a schedule with only one critical path, the makespan is reduced if and only if the swap of operations $u$ and $v$ violates both Propositions 2 and 3. This case is covered by Theorem 1, proven in Appendix D, although we do not explicitly use this theorem in our policy network. In schedules with multiple critical paths, the swap of operations $u$ and $v$ that violates all three Propositions 1, 2, and 3 either reduces the makespan or the number of critical paths, as proven in Theorem 2 and Theorem 3 (Appendix D).

**Theorem 1.** *For a schedule with only one critical path, the makespan of a given schedule decreases when consecutive operations $u$ and $v$ are swapped on a machine if and only if $lst_{JS[u]} > ect'_u$ and $lst_{MS[v]} > ect'_u$.*

**Theorem 2.** *For a schedule with multiple critical paths, the makespan of a given schedule decreases when consecutive operations $u$ and $v$ are swapped on a machine if and only if $lst_{JS[u]} > ect'_u$, $lst_{MS[v]} > ect'_u$, and all critical paths include consecutive operations $u$ and $v$.*

**Theorem 3.** *For a schedule with multiple critical paths, the makespan or the number of critical paths of the schedule decreases when consecutive operations $u$ and $v$ on a machine are swapped, if $lst_{JS[v]} \leq ect'_v$, $lst_{JS[u]} > ect'_u$, and $lst_{MS[v]} > ect'_u$.*

**Embedding structure.** For a candidate action involving the swap of operations $u$ and $v$, the decoder generates a joint embedding vector $h_{uv}$ defined as $[h_u \| x_u \| h_v \| x_v \| h_G \| \mathbb{1}_{P(u,v)} \| \mathbb{1}_{T(u,v)}]$, where $\|$ represents concatenation, $h_u$ is the embedding vector of operation $u$, $x_u$ is the state feature of operation $u$, $h_G$ is the average embedding vector across all operations, $\mathbb{1}_{P(u,v)}$ denotes a set of binary indicators representing whether the swap of $(u,v)$ satisfies the conditions of three propositions we have developed, and $\mathbb{1}_{T(u,v)}$ indicates whether the swap action is in the list of recently swapped pairs. $h_{uv}$ is fed into an MLP, whose outputs are normalized via softmax to yield selection probabilities over all candidate swaps. At each step, the swap with the highest probability is selected as the next move.

**Adaptive list size of recently swapped pairs.** LSI adopts an adaptive tabu strategy (Zhang et al. (2007)), which is also used in TBGAT. In contrast to conventional tabu search that excludes recently swapped pairs, LSI retains all candidate actions even though be recently swapped, and encodes their revisit status as a binary feature, allowing the policy to learn when revisiting such moves is beneficial. This approach is implemented in TBGAT's publicly available code, although it is not explicitly documented in the original paper. The size of the list is adaptively determined based on the ratio of the number of jobs to the number of machines in each JSSP instance, with additional stochastic variation, as detailed in Appendix E.

## 5 LEARNING PROCESS

Our model is trained via RL, specifically an entropy-regularized $n$-step REINFORCE algorithm, as adopted in TBGAT (Zhang et al. (2024b)). The entropy term promotes generalization and encourages exploration of the action space. At each time step $t$, we compute the normalized, discounted cumulative reward $\bar{R}_t$ and use it to weight the log-likelihood of the chosen action under our policy $\pi_\theta(a_t | s_t)$. We add an entropy $\mathcal{H}(\pi_\theta) = -\mathbb{E}_{a \sim \pi_\theta} \log(\pi_\theta(a))$, scaled by a factor $\beta$ which controls the strength of entropy regularization, to encourage exploration. Concretely, we minimize the loss:

$$\mathcal{L}(\theta) = -\sum_t [\bar{R}_t \log \pi_\theta(a_t | s_t) + \beta \mathcal{H}(\pi_\theta(\cdot | s_t))]. \tag{1}$$

Table 1: Mean optimality gap for each JSSP benchmark group across different methods.

| Method | TA 15x15 Gap | Time | TA 20x15 Gap | Time | TA 20x20 Gap | Time | TA 30x15 Gap | Time | TA 30x20 Gap | Time | TA 50x15 Gap | Time | TA 50x20 Gap | Time | TA 100x20 Gap | Time | ABZ 10x10 Gap | Time | ABZ 20x15 Gap | Time | FT 6x6 Gap | Time | FT 10x10 Gap | Time | FT 20x5 Gap | Time |
|---|---|---|---|---|---|---|---|---|---|---|---|---|---|---|---|---|---|---|---|---|---|---|---|---|---|---|
| CP (10 sec) | 0.3% | 9.4s | 3.2% | 9.3s | 3.8% | 10.1s | 8.5% | 9.2s | 13.2% | 10.0s | 10.2% | 9.8s | 15.7% | 9.8s | 12.5% | 9.1s | 0.0% | 0.5s | 3.1% | 10.1s | 0.0% | 3.1s | 0.0% | 0.6s | 0.0% | 0.0s |
| CP (1 min) | 0.2% | 42.4s | 1.4% | 50.9s | 1.8% | 1.0m | 4.2% | 54.4s | 6.0% | 1.0m | 3.6% | 1.0m | 6.9% | 1.0m | 8.8% | 59.1s | 0.0% | 0.7s | 1.9% | 1.0m | 0.0% | 3.2s | 0.0% | 0.5s | 0.0% | 0.0s |
| CP (5 min) | 0.0% | 2.2m | 1.2% | 4.3m | 1.6% | 5.0m | 2.4% | 4.5m | 4.0% | 5.0m | 0.2% | 3.3m | 2.1% | 4.8m | 4.9% | 5.0m | 0.0% | 0.8s | 0.8% | 5.0m | 0.0% | 7.2s | 0.0% | 0.5s | 0.0% | 0.0s |
| CP (30 min) | 0.0% | 5.2m | 0.7% | 24.1m | 0.7% | 30.0m | 1.1% | 21.6m | 2.6% | 30.0m | 0.0% | 2.0m | 0.3% | 16.8m | 0.0% | 12.5m | 0.0% | 0.5s | 0.4% | 21.0m | 0.0% | 2.0s | 0.0% | 0.3s | 0.0% | 0.0s |
| MOR | 16.7% | 0.2s | 23.4% | 0.3s | 19.5% | 0.3s | 21.5% | 0.4s | 24.0% | 0.7s | 15.6% | 0.9s | 16.2% | 1.3s | 7.7% | 3.2s | 7.6% | 0.1s | 17.5% | 0.3s | 23.6% | 0.1s | 26.1% | 0.1s | 34.3% | 0.0s |
| LTT | 19.5% | 0.2s | 20.0% | 0.3s | 18.4% | 0.3s | 20.3% | 0.4s | 23.7% | 0.7s | 14.3% | 0.9s | 15.2% | 1.3s | 6.5% | 3.2s | 6.5% | 0.1s | 19.5% | 0.3s | 9.1% | 0.1s | 17.2% | 0.1s | 24.8% | 0.0s |
| FDD/MWKR | 17.7% | 0.2s | 21.3% | 0.3s | 19.9% | 0.3s | 21.7% | 0.4s | 24.0% | 0.7s | 15.3% | 0.9s | 16.3% | 1.3s | 7.7% | 3.2s | 9.7% | 0.1s | 17.9% | 0.3s | 21.8% | 0.1s | 18.0% | 0.1s | 27.0% | 0.0s |
| SN | 15.3% | 3.5s | 19.4% | 6.6s | 17.2% | 11.0s | 19.1% | 17.1s | 23.7% | 28.3s | 13.9% | 52.5s | 13.5% | 1.6m | 6.7% | 7.4m | 6.1% | 0.7s | 20.5% | 6.6s | 7.3% | 0.8s | 19.5% | 1.6s | 28.6% | 0.2s |
| IRD | 8.9% | 2.5s | 11.7% | 3.8s | 12.5% | 4.4s | 11.6% | 7.2s | 14.4% | 8.7s | 4.9% | 15.7s | 9.5% | 20.3s | 2.3% | 1.3m | 4.8% | 1.0s | 10.7% | 3.5s | 5.5% | 1.0s | 7.1% | 1.5s | 3.3% | 0.4s |
| NeuroLS-500 | 6.2% | 6.8s | 9.9% | 8.7s | 9.8% | 10.9s | 12.0% | 12.9s | 14.6% | 16.3s | 9.5% | 21.9s | 9.9% | 27.9s | 5.0% | 1.0m | 1.5% | 4.1s | 10.5% | 9.0s | **0.0%** | 2.8s | 2.4% | 4.2s | 9.6% | 4.7s |
| NeuroLS-1000 | 4.9% | 13.7s | 8.5% | 17.4s | 8.5% | 21.7s | 10.9% | 25.8s | 13.0% | 32.6s | 8.1% | 43.9s | 8.9% | 55.8s | 4.3% | 2.0m | 1.5% | 8.2s | 9.6% | 18.1s | **0.0%** | 5.5s | 2.4% | 8.3s | 3.4% | 9.4s |
| NeuroLS-5000 | **3.1%** | 1.1m | 5.7% | 1.5m | 5.3% | 1.8m | 6.8% | 2.1m | 8.9% | 2.7m | 3.4% | 3.7m | 5.2% | 4.7m | 2.0% | 10.0m | 1.1% | 40.8s | **5.1%** | 1.5m | **0.0%** | 27.6s | 2.3% | 41.5s | 2.2% | 47.2s |
| L2S-500 | 8.8% | 9.3s | 11.9% | 10.1s | 12.0% | 10.9s | 15.4% | 12.7s | 18.7% | 14.0s | 11.1% | 16.2s | 13.7% | 22.8s | 8.0% | 50.2s | 2.8% | 7.4s | 13.5% | 10.2s | 3.6% | 6.8s | 9.9% | 7.5s | 7.0% | 7.4s |
| L2S-1000 | 6.3% | 18.7s | 10.5% | 20.3s | 11.2% | 22.2s | 13.4% | 24.7s | 16.7% | 28.4s | 8.9% | 32.9s | 11.6% | 45.4s | 6.2% | 1.7m | 2.8% | 15.0s | 11.9% | 19.9s | **0.0%** | 13.5s | 8.0% | 15.1s | 7.0% | 15.0s |
| L2S-5000 | 5.5% | 1.4m | 8.6% | 1.7m | 8.8% | 1.9m | 9.5% | 2.0m | 12.7% | 2.4m | 7.1% | 2.8m | 7.1% | 3.8m | 2.3% | 8.4m | 1.4% | 1.3m | 8.9% | 1.7m | **0.0%** | 1.1m | 5.7% | 1.2m | 3.9% | 1.2m |
| TBGAT-500 | 7.9% | 12.6s | 10.4% | 14.6s | 11.3% | 17.5s | 15.7% | 17.2s | 18.1% | 19.3s | 11.0% | 23.9s | 12.3% | 24.4s | 7.1% | 42.0s | 1.1% | 9.2s | 10.0% | 12.8s | **0.0%** | 7.4s | 5.2% | 10.3s | 9.5% | 11.7s |
| TBGAT-1000 | 6.5% | 24.9s | 8.8% | 28.7s | 9.8% | 34.1s | 13.7% | 33.7s | 15.5% | 37.3s | 8.9% | 46.9s | 10.4% | 47.5s | 5.6% | 1.4m | 1.1% | 17.9s | 9.7% | 25.3s | **0.0%** | 14.2s | 4.8% | 20.5s | 6.7% | 23.2s |
| TBGAT-5000 | 4.8% | 2.1m | 7.1% | 2.3m | 7.4% | 2.7m | 10.3% | 2.7m | 11.2% | 2.9m | 5.0% | 3.9m | 6.1% | 3.9m | 2.0% | 6.7m | 0.8% | 1.5m | 6.5% | 2.1m | **0.0%** | 1.2m | 2.9% | 1.7m | 4.0% | 1.9m |
| LSI-500 | 6.3% | 16.5s | 8.6% | 16.1s | 9.4% | 17.5s | 12.1% | 19.5s | 14.8% | 22.8s | 8.4% | 24.9s | 10.1% | 28.1s | 5.7% | 46.0s | 2.5% | 8.2s | 9.8% | 18.0s | 5.5% | 6.0s | 9.5% | 9.1s | **2.2%** | 10.4s |
| LSI-1000 | 5.2% | 33.9s | 7.4% | 32.1s | 8.0% | 35.3s | 9.9% | 37.9s | 12.2% | 45.9s | 6.3% | 49.8s | 8.1% | 54.3s | 3.9% | 1.5m | 1.4% | 16.6s | 8.8% | 36.5s | **0.0%** | 11.7s | 6.0% | 18.2s | **2.2%** | 20.5s |
| LSI-5000 | 3.6% | 2.7m | **5.0%** | 2.6m | **5.0%** | 2.9m | **6.1%** | 3.4m | **8.4%** | 3.6m | **2.0%** | 4.2m | **4.8%** | 4.6m | **0.9%** | 7.7m | **0.8%** | 1.4m | 5.4% | 2.4m | **0.0%** | 58.0s | **2.0%** | 1.5m | **2.2%** | 1.8m |

| Method | LA 10x5 Gap | Time | LA 15x5 Gap | Time | LA 20x5 Gap | Time | LA 10x10 Gap | Time | LA 15x10 Gap | Time | LA 20x10 Gap | Time | LA 30x10 Gap | Time | LA 15x15 Gap | Time | SWV 20x10 Gap | Time | SWV 20x15 Gap | Time | SWV 50x10 Gap | Time | ORB 10x10 Gap | Time | YN 20x20 Gap | Time |
|---|---|---|---|---|---|---|---|---|---|---|---|---|---|---|---|---|---|---|---|---|---|---|---|---|---|---|
| CP (10 sec) | 0.0% | 0.2s | 0.0% | 0.0s | 0.0% | 6.2s | 0.4% | 4.6s | 0.0% | 0.1s | 0.5% | 4.9s | 0.0% | 0.1s | 0.0% | 2.3s | 2.4% | 10.0s | 7.0% | 10.0s | 6.1% | 6.8s | 0.0% | 2.6s | 4.3% | 10.0s |
| CP (1 min) | 0.0% | 0.2s | 0.0% | 0.0s | 0.0% | 17.5s | 0.0% | 14.6s | 0.0% | 0.1s | 0.3% | 18.7s | 0.0% | 0.1s | 0.0% | 2.0s | 2.0% | 50.2s | 3.4% | 1.0m | 3.8% | 32.4s | 0.0% | 2.8s | 1.7% | 1.0m |
| CP (5 min) | 0.0% | 0.5s | 0.0% | 0.0s | 0.0% | 28.9s | 0.0% | 56.8s | 0.0% | 0.1s | 0.4% | 1.1m | 0.0% | 0.2s | 0.0% | 3.4s | 1.8% | 4.5m | 4.2% | 5.0m | 2.2% | 2.6m | 0.0% | 4.4s | 1.8% | 5.0m |
| CP (30 min) | 0.0% | 0.2s | 0.0% | 0.1s | 0.0% | 14.6s | 0.0% | 22.5s | 0.0% | 0.1s | 0.0% | 6.0m | 0.0% | 0.1s | 0.0% | 1.9s | 0.8% | 19.8m | 3.1% | 30.0m | 0.6% | 7.8m | 0.0% | 2.4s | 1.3% | 30.0m |
| MOR | 13.0% | 0.1s | 2.4% | 0.0s | 2.9% | 0.1s | 13.3% | 0.2s | 21.0% | 0.1s | 17.4% | 0.2s | 6.3% | 0.1s | 18.2% | 0.3s | 40.9% | 0.2s | 35.0% | 0.3s | 29.1% | 0.5s | 26.6% | 0.1s | 18.1% | 0.4s |
| LTT | 12.6% | 0.1s | 2.3% | 0.0s | 3.5% | 0.1s | 13.0% | 0.2s | 15.3% | 0.1s | 16.5% | 0.2s | 4.1% | 0.1s | 15.1% | 0.3s | 29.3% | 0.2s | 29.3% | 0.3s | 21.5% | 0.5s | 19.8% | 0.1s | 19.7% | 0.4s |
| FDD/MWKR | 13.2% | 0.1s | 4.6% | 0.0s | 2.8% | 0.1s | 14.9% | 0.2s | 13.1% | 0.1s | 19.3% | 0.2s | 7.8% | 0.1s | 16.6% | 0.3s | 38.3% | 0.2s | 34.5% | 0.3s | 23.5% | 0.5s | 19.8% | 0.1s | 20.9% | 0.4s |
| SN | 12.1% | 0.6s | 2.7% | 1.2s | 3.6% | 1.9s | 11.9% | 0.8s | 14.6% | 2.0s | 15.7% | 4.1s | 3.1% | 9.3s | 16.1% | 3.5s | 34.4% | 3.9s | 30.6% | 6.7s | 25.4% | 25.1s | 20.0% | 0.8s | 18.4% | 11.2s |
| IRD | 4.6% | 0.9s | 0.1% | 0.6s | 0.1% | 1.8s | 4.1% | 2.2s | 8.5% | 1.1s | 5.9% | 2.8s | 0.9% | 1.6s | 10.8% | 4.9s | 11.8% | 2.8s | **13.0%** | 3.8s | **4.2%** | 9.7s | 9.0% | 1.0s | 13.9% | 4.3s |
| NeuroLS-500 | 0.9% | 3.0s | **0.0%** | 4.0s | **0.0%** | 5.0s | 2.8% | 3.9s | 3.9% | 5.6s | 5.4% | 7.3s | 0.1% | 10.8s | 5.7% | 7.1s | 24.3% | 7.4s | 22.3% | 8.7s | 20.6% | 16.3s | 5.3% | 3.8s | 9.4% | 10.7s |
| NeuroLS-1000 | 0.9% | 6.0s | **0.0%** | 8.1s | **0.0%** | 10.0s | 2.4% | 7.8s | 3.7% | 11.2s | 4.0% | 14.5s | 0.1% | 21.6s | 4.9% | 14.3s | 22.1% | 14.8s | 20.2% | 17.5s | 19.6% | 32.7s | 3.9% | 7.6s | 7.8% | 21.3s |
| NeuroLS-5000 | **0.0%** | 29.8s | **0.0%** | 40.4s | **0.0%** | 50.2s | 1.1% | 39.2s | **2.3%** | 55.9s | 2.2% | 1.2m | **0.0%** | 1.8m | 2.7% | 1.2m | 12.5% | 1.2m | 14.9% | 1.5m | 16.4% | 2.7m | **1.9%** | 38.0s | 5.2% | 1.8m |
| L2S-500 | 2.1% | 6.9s | **0.0%** | 6.8s | **0.0%** | 7.1s | 4.4% | 7.5s | 5.4% | 8.0s | 6.9% | 8.9s | 0.1% | 10.2s | 7.4% | 9.0s | 27.7% | 8.8s | 26.5% | 9.7s | 21.4% | 12.5s | 8.2% | 7.4s | 13.9% | 11.7s |
| L2S-1000 | 1.8% | 14.0s | **0.0%** | 13.9s | **0.0%** | 14.5s | 2.3% | 15.0s | 4.8% | 16.0s | 6.4% | 17.5s | **0.0%** | 20.4s | 7.2% | 18.2s | 25.1% | 17.6s | 24.2% | 19.0s | 19.9% | 25.4s | 6.6% | 15.0s | 11.5% | 23.4s |
| L2S-5000 | 1.8% | 1.2m | **0.0%** | 1.2m | **0.0%** | 1.2m | **0.9%** | 1.3m | 4.2% | 1.4m | 4.0% | 1.4m | **0.0%** | 1.7m | 5.4% | 1.5m | 19.9% | 1.4m | 18.2% | 1.7m | 17.3% | 2.1m | 3.8% | 1.3m | 8.8% | 1.9m |
| TBGAT-500 | 3.5% | 2.3s | 1.1% | 0.9s | 0.1% | 1.7s | 1.8% | 9.1s | 5.8% | 10.8s | 6.7% | 11.4s | 1.5% | 4.9s | 7.0% | 12.1s | 31.0% | 15.6s | 24.3% | 17.0s | 21.1% | 29.8s | 7.0% | 10.4s | 10.4% | 14.3s |
| TBGAT-1000 | 3.5% | 3.1s | 1.1% | 1.9s | **0.0%** | 3.3s | 1.8% | 18.2s | 5.4% | 21.4s | 5.4% | 22.8s | 1.4% | 6.3s | 4.8% | 24.5s | 29.8% | 31.2s | 23.0% | 34.3s | 20.1% | 59.5s | 5.7% | 20.8s | 8.5% | 28.1s |
| TBGAT-5000 | 3.5% | 9.8s | **0.0%** | 9.2s | **0.0%** | 16.1s | 1.4% | 1.6m | 2.8% | 1.8m | 4.9% | 52.0s | 1.1% | 16.9s | 3.8% | 2.0m | 27.7% | 2.5m | 17.0% | 2.7m | 17.6% | 4.9m | 4.5% | 1.7m | 5.6% | 2.3m |
| LSI-500 | 2.9% | 9.0s | **0.0%** | 9.1s | **0.0%** | 3.8s | 2.0% | 12.1s | 4.9% | 11.5s | 5.6% | 13.4s | **0.0%** | 10.5s | 5.1% | 14.0s | 25.4% | 21.6s | 21.4% | 19.7s | 20.7% | 42.2s | 5.3% | 13.5s | 9.2% | 15.4s |
| LSI-1000 | 2.3% | 18.1s | **0.0%** | 17.9s | **0.0%** | 5.3s | 1.9% | 23.5s | 4.4% | 22.7s | 3.9% | 26.0s | **0.0%** | 11.8s | 3.8% | 27.8s | 21.8% | 41.3s | 18.6% | 37.8s | 20.4% | 1.2m | 4.6% | 26.1s | 7.3% | 30.4s |
| LSI-5000 | 1.1% | 1.4m | **0.0%** | 1.5m | **0.0%** | 17.7s | 0.9% | 1.9m | 3.2% | 2.0m | **1.8%** | 2.1m | **0.0%** | 22.1s | **2.4%** | 2.2m | **11.2%** | 2.8m | 13.5% | 3.1m | 18.9% | 5.4m | 2.2% | 2.3m | **5.0%** | 2.5m |

Policy parameters are updated periodically based on gradient estimates gathered from sample trajectories. The entire learning process, which includes trajectory collection, gradient computation, and parameter updates, is provided in Algorithm 1 in Appendix F.

## 6 EXPERIMENTS

**Baselines and test datasets.** We compared the performance of our method with dispatching rule-based constructive heuristic methods (largest tail time (LTT) rule (Lee & Kim (2024)), most operations remaining (MOR) rule, minimum ratio of flow due date to most work remaining (FDD/MWKR) rule (Sels et al. (2012))), learning-based constructive heuristics (SN (Park et al. (2021a)) and IRD (Lee & Kim (2024))), local search methods (TS Zhang et al. (2007), NeuroLS (Falkner et al. (2022)), L2S (Zhang et al. (2024a)), and TBGAT (Zhang et al. (2024b))), and an exact method (constraint programming (CP) (Zhou (1996))). We evaluated all methods on JSSP benchmark datasets: TA (Taillard (1993)), LA (Lawrence (1984)), ABZ (Adams et al. (1988)), FT (Muth & Thompson (1963)), ORB (Applegate & Cook (1991)), SWV (Storer et al. (1992)), and YN (Yamada & Nakano (1992)). An instance denoted as 'benchmark $N$x$M$' contains $N$ jobs and $M$ machines. Performance is measured by the optimality gap, defined as $\frac{C}{C^*} - 1$, where $C$ is the makespan of a schedule obtained by each method and $C^*$ is the optimal or best-known makespan.

**Performance for JSSP benchmark datasets.** We initialized the schedules using the FDD/MWKR rule and computed EST and LST using the topological linear-time algorithm proposed by Zhang et al. Zhang et al. (2024a). L2S, TBGAT, and LSI were trained and validated on JSSP instances with 10 jobs and 10 machines, while NeuroLS used instances with 15 jobs and 15 machines. Details of implementation and configuration are provided in Appendix G.

Table 1 reports the results grouped by benchmark type, problem size, and method. The number following each learning-based local search method (NeuroLS, L2S, TBGAT, and LSI) in the table denotes the number of search iterations, and the reported running time is derived by batching computations across instances within each instance group. Although LSI is trained only on 10x10 instances, it generalizes well to significantly larger instances, highlighting its strong scalability. LSI also demonstrates superior performance compared to other learning-based methods while requiring similar computational resources. In particular, LSI outperforms both L2S and TBGAT across nearly all benchmark groups with the same search iterations. Figure 5 also shows the performance and computational time of large-scale JSSP benchmarks. In terms of both the solution quality (optimality gap) and the computational efficiency (end-to-end running time), LSI outperforms all learning-based methods, classical dispatchers, and the CP solver.

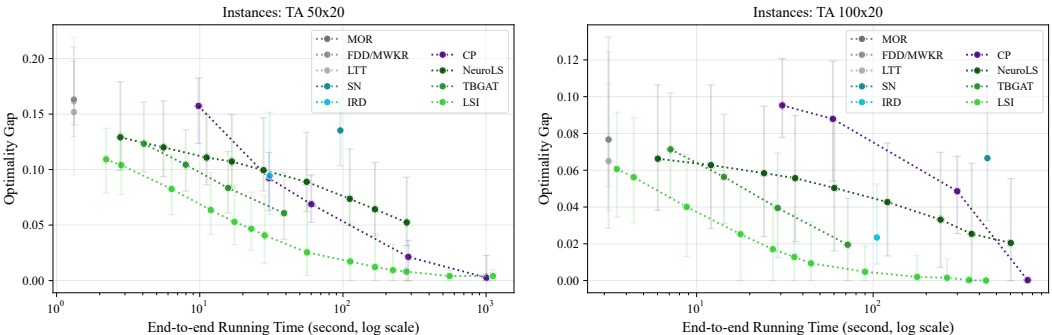

Figure 5: Mean optimality gap and mean end-to-end running time for large-scale benchmarks.

**Ablation studies.** Table 2 reports the performance of different policy network architectures. In the table, 'Gap' denotes the average optimality gap across all 162 JSSP benchmark instances, 'Rank' indicates the average rank among the four compared methods within each iteration, and 'Diff' represents the average difference in optimality gap from the best-performing method for each instance with the same number of iterations. A 'Diff' of 0% indicates that the method consistently achieved the best performance across all instances under its iteration setting. The inclusion of condition indicators significantly improves performance for both encoder types, a simple MLP and TBGAT's

Table 2: Evaluation of different policy network architecture under varying search iterations.

| # of iterations | 500 | | | 1000 | | | 5000 | | |
|---|---|---|---|---|---|---|---|---|---|
| Method | Gap | Rank | Diff | Gap | Rank | Diff | Gap | Rank | Diff |
| None + MLP | 12.7% | 3.60 | 4.3% | 11.2% | 3.50 | 4.3% | 8.2% | 3.35 | 3.9% |
| None + bi-GAT (TBGAT) | 10.7% | 2.78 | 2.4% | 9.4% | 2.78 | 2.5% | 7.0% | 2.81 | 2.6% |
| Indicators + MLP (LSI) | 9.0% | 1.86 | 0.7% | **7.6%** | **1.86** | **0.6%** | **5.0%** | **1.85** | **0.6%** |
| Indicators + bi-GAT | **8.9%** | **1.76** | **0.6%** | 7.6% | 1.86 | 0.6% | 5.2% | 1.99 | 0.8% |

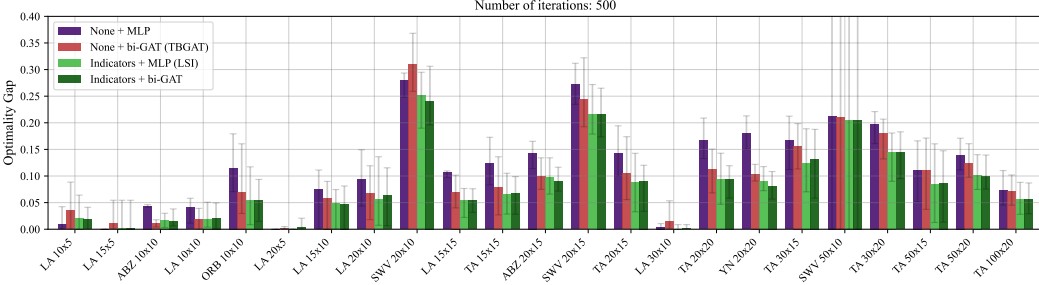

Figure 6: Mean optimality gap for benchmark groups with different policy network structures.

bi-GAT. 'Indicators' in Table 2 refers to the use of our proposed condition indicators in the decoder, while 'None' indicates that no theoretical insights are incorporated into the policy network. Both encoders with condition indicators achieve nearly identical performance, differing by less than 0.2%. This suggests that our theoretically derived indicators can serve as an effective substitute for complex neural architectures designed to capture problem-specific features. Figure 6 also shows the effectiveness of the condition indicators across different benchmark groups. As shown in the figure, within each benchmark group of the same size, the performance difference between using a simple MLP and a bi-GAT encoder remains negligible when the condition indicators are used. The other results of ablation studies across other four components are presented in Appendix I: the effectiveness of (1) different ways to incorporate the propositions into the local search method, (2) different state features, (3) different critical-based neighborhood structures, and (4) different training instance sizes.

## 7 CONCLUSION

We propose **LSI** (Local Search with Indicators), a novel learning-based local search method for makespan minimization in Job Shop Scheduling Problems (JSSPs). LSI replaces complex GNN architectures with a lightweight MLP policy network that incorporates three binary indicators derived from newly identified necessary conditions for makespan reduction. These conditions, which we define through a theoretical analysis of schedule improvement under consecutive operation swaps, are used to guide the policy network toward selecting only the most promising moves within the $N5$ neighbors. By embedding these problem-specific theoretical insights into the policy, LSI achieves superior performance and scalability across JSSP benchmarks while significantly reducing inference time. Experimental results across diverse JSSP benchmarks demonstrate that our proposed indicators not only enhance the solution quality but also enable the use of a lightweight MLP-based encoder, outperforming prior methods that rely on complex neural architectures such as Graph Attention Network (GAT). This result highlights that problem-specific insights related to the objective function can serve as an effective substitute for architectural complexity.

**Limitations and future works.** While the approach shows strong performance, it has some limitations. The theoretical indicators we propose are problem-specific and must be manually derived for each combinatorial optimization problem (COP), which may limit generalizability and automation of this approach. Future work will focus on two directions. One is to generalize our method to other COPs, such as vehicle routing problems or scheduling problems with additional constraints, by deriving analogous theoretical indicators. The other is to develop representation learning techniques that can autonomously discover and learn such objective-aligned features, reducing reliance on hand-crafted indicators.

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

ACKNOWLEDGMENTS

This work was supported by (We will add a funding program after accept), and this paper have edited by using several LLM services.

APPENDIX

We provide further details of our paper in the appendix. Our code implementation can be found in `https://github.com/***/***` (Code will be made publicly available upon acceptance).

## A  DISJUNCTIVE GRAPH TO REPRESENT JSSP INSTANCE

Although our proposed method, LSI, does not directly employ a disjunctive graph-based representation, we introduce it here to provide context for comparison with prior works that have utilized this structure to learn architectural information implied in the JSSP.

The JSSP can be modeled as a disjunctive graph (Błażewicz et al. (2000)), as illustrated in Figure 7. In this representation, each operation—including dummy operations—is represented as a node. Specifically, each node $O_{ij}$ denotes the $j$-th operation of job $i$, which must be processed on a specified machine for a given processing time. Dummy nodes $O_S$ and $O_T$ represent the artificial start and terminal operations with zero processing time. Nodes with the same color belong to the same job, indicating their precedence relationship.

There are two types of arcs in the disjunctive graph: conjunctive (directed) arcs represent precedence constraints between successive operations within a job, while disjunctive (undirected) arcs connect operations assigned to the same machine. Once the processing sequence between operations assigned to the same machine is determined, the corresponding disjunctive arc becomes directed.

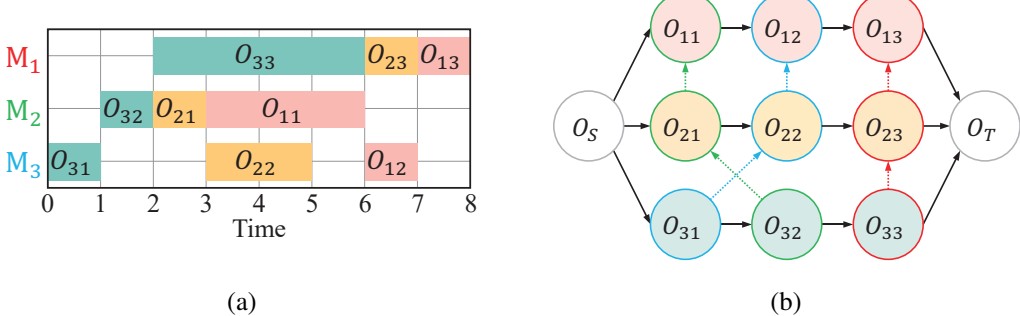

(a)                          (b)

Figure 7: (a) A schedule of a JSSP instance with three jobs and three machines and (b) the disjunctive graph of the schedule.

Several prior studies have adopted GNN architectures based on the disjunctive graph-based representation to encode operations' features (Zhang et al. (2020); Park et al. (2021b); Liu & Huang (2023); Park et al. (2021a); Lee & Kim (2022; 2024); Falkner et al. (2022); Zhang et al. (2024a;b)). This representation captures both the precedence relations across operations within a job and the operation sequence on each machine under the current schedule. Such structural encoding enhances the model's capacity to capture scheduling-specific characteristics and the underlying topological information of the current schedule.

## B  CRITICAL PATH-BASED NEIGHBORHOOD STRUCTURES

As described in Section 2, a critical path is defined as the longest path from the operation with the earliest start time to the operation with the latest completion time, where the total path length equals the sum of processing times of the operations along the path. No idle time exists between consecutive operations on the critical path, and its length is equal to the schedule's makespan. For

instance, Figure 8 shows all critical paths that can be found in the schedule illustrated in Figure 7. The critical block is a subset of the operations on the critical path and is defined as maximal sequences of consecutive operations processed on the same machine within the critical path. For example, there are three critical blocks $\{O_{31}\}$, $\{O_{32}\}$, and $\{O_{33}, O_{23}, O_{13}\}$ on the critical path of Figure 8 (a), and four critical blocks $\{O_{31}\}$, $\{O_{32}, O_{21}, O_{11}\}$, $\{O_{12}\}$, and $\{O_{13}\}$ on the critical path of Figure 8 (b).

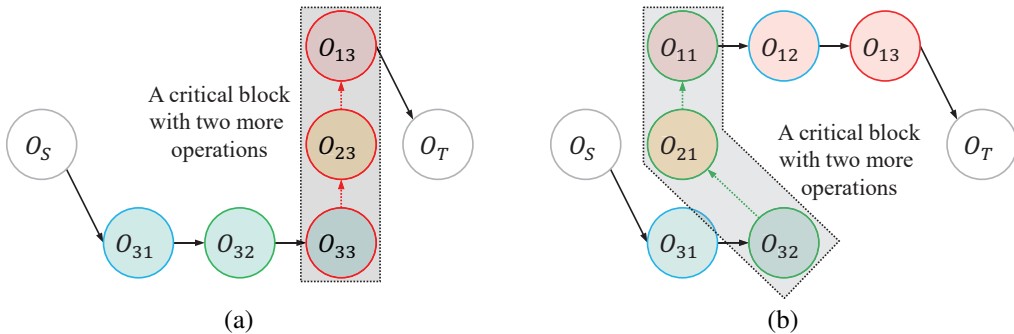

(a)  (b)

Figure 8: Two critical paths of the schedule illustrated in Figure 7.

A neighborhood structure generates neighbor schedules from the current schedule in local search methods. The $N1$ neighborhood structure considers all possible swaps of consecutive operations in critical blocks (Van Laarhoven et al. (1992)). $N2$ narrows these to pairs at the beginning or end of critical blocks (Dell'Amico & Trubian (1993)). $N5$ further refines $N2$ by excluding pairs at the beginning of the first critical block or the end of the last, except when these are at the end of the first block or the beginning of the last, respectively (Nowicki & Smutnicki (1996)). For example, in the critical paths shown in Figures 8 (a) and (b), $N5$ considers the pairs $(O_{33}, O_{23})$, $(O_{23}, O_{21})$, and $(O_{21}, O_{11})$ as candidates. The pair $(O_{23}, O_{13})$ is excluded because $O_{13}$ is the last operation in the critical path and $O_{23}$ is not at the beginning of its block.

All three neighborhood structures ensure feasibility by generating acyclic disjunctive graphs only (Van Laarhoven et al. (1992)). While $N1$ can theoretically reach the optimal schedule from any initial solution (Van Laarhoven et al. (1992)), $N5$ is widely adopted for its efficiency. It considers fewer operation pairs than $N1$ and $N2$ while including all pairs whose swaps can potentially improve the makespan Nowicki & Smutnicki (1996), as shown in Figure 2. This makes $N5$ effective in practice.

## C    PROOFS OF NECESSARY CONDITIONS FOR MAKESPAN REDUCTION

The proofs of three propositions proposed in Section 4.3 are as follows. $JP[u]$ and $JS[u]$ denote the job-predecessor and job-successor operations of operation $u$, respectively, while $MP[u]$ and $MS[u]$ represent its machine-predecessor and machine-successor operations, respectively. Note that $ect'_v$ can be calculated as $\max(ect_{MP[u]}, ect_{JP[v]}) + p_v$ and $ect'_u$ can be calculated as $\max(est_u, ect'_v) + p_u$. The propositions hold for all consecutive operations on the same machine, not just those on critical paths.

**Proposition 1.**    *When consecutive operations $u$ and $v$ are swapped on a machine of a given schedule, the makespan cannot decrease if $lst_{JS[v]} \leq ect'_v$.*

*Proof.* By definition, LST is the latest time an operation can start without delaying the makespan. Therefore, increasing the LST of any job's last operation increases the makespan. If $lst_{JS[v]} \leq ect'_v$, then $lst_{JS[v]} \leq ect'_v \leq est'_{JS[v]} \leq lst'_{JS[v]}$ due to the precedence constraint of $v$ and the definitions of EST and LST. From the perspective of the job of $v$, the LST of its subsequent operations, including $JS[v]$, will remain the same or be delayed. This implies that the LST of the last operation of the job of $v$ will also remain the same or be delayed. Consequently, the makespan will either remain the same or increase. □

**Proposition 2.**    *When consecutive operations $u$ and $v$ are swapped on a machine of a given schedule, the makespan cannot decrease if $lst_{JS[u]} \leq ect'_u$.*

*Proof.* The proof follows the same logic as Proposition 1. The condition $lst_{JS[u]} \leq ect'_u$ implies $lst_{JS[u]} \leq lst'_{JS[u]}$, which ensures that the makespan cannot decrease. ☐

**Proposition 3.** *When consecutive operations $u$ and $v$ are swapped on a machine of a given schedule, the makespan cannot decrease if $lst_{MS[v]} \leq ect'_u$.*

*Proof.* We consider determined operation orders for all machines except for operations $u$ and $v$. By the definition of LST, increasing the LST of the last operation processed on any machine increases the makespan. If $lst_{MS[v]} \leq ect'_u$, then $lst_{MS[v]} \leq ect'_u \leq est'_{MS[v]} \leq lst'_{MS[v]}$ since operation $u$ precedes $MS[v]$ on their compatible machine. From the machine's perspective, the LST of subsequent operations including $MS[v]$ will either remain the same or be delayed. This implies that the LST of the last operation on the machine will also either remain the same or be delayed. Consequently, the makespan will either remain the same or increase. ☐

# D   PROOF OF SUFFICIENT CONDITION FOR MAKESPAN REDUCTION

**Definitnion: Downstream-affected operations.**   Given a swap of consecutive operations $(u, v)$ on a machine, let $F^+(u, v)$ denote the set of operations reachable from $u$ or $v$ by repeatedly taking a job successor or a machine successor. We call elements of $F^+(u, v)$ *downstream-affected operations*. Note that $u$ and $v$ are not in $F^+(u, v)$.

**Lemma 1.** *Locality of perturbation. Swapping $(u, v)$ can change EST and ECT only for downstream-affected operations: for any operation $w \notin F^+(u, v)$, $est'_w = est_w$ and $ect'_w = ect_w$.*

*Proof.* Suppose $est'_w \neq est_w$ for some $w \notin F^+(u, v)$. Since $est_w = \max\{ect_{JP[w]}, ect_{MP[w]}\}$, a change at $w$ implies $ect'_x \neq ect_x$ for some immediate predecessor $x \in \{JP[w], MP[w]\}$. Iterating the same reasoning on $x$ (and so on) produces a finite backward chain of nodes with changed $est/ect$ that must originate at $u$ or $v$, the only place where the schedule was modified. Hence, there exists a successor sequence from $u$ or $v$ to $w$. This is in contradiction with $w \notin F^+(u, v)$. Therefore, $est'_w = est_w$ and thus $ect'_w = ect_w$ for all $w \notin F^+(u, v)$. ☐

**Lemma 2.** *LST monotonicity for downstream-affected operations. Let $O_T$ denote the terminal dummy operation. If $lst'_{O_T} \geq lst_{O_T}$ (the makespan is the same or increasing), then we have $lst'_w \geq lst_w$ for any $w \in F^+(u, v)$.*

*Proof.* For each operation $x$, $lst_x = \min\{lst_{JS[x]}, lst_{MS[x]}\} - p_x$. Order the nodes of $F^+(u, v)$ by their successor distance to $O_T$ and proceed by induction.

If $w$'s only successor is $O_T$, then $lst'_w = lst'_{O_T} - p_w \geq lst_{O_T} - p_w = lst_w$. If $lst'_{JS[w]} \geq lst_{JS[w]}$ and $lst'_{MS[w]} \geq lst_{MS[w]}$, then $lst'_w = \min\{lst'_{JS[x]}, lst'_{MS[x]}\} - p_w \geq lst_w = \min\{lst_{JS[x]}, lst_{MS[x]}\} - p_w$. Therefore, $lst'_w \geq lst_w$ for all $w \in F^+(u, v)$ by induction. ☐

**Theorem 1**   For a schedule with only one critical path, the makespan of the schedule decreases when consecutive operations $u$ and $v$ on a machine are swapped if and only if $lst_{JS[u]} > ect'_u$ and $lst_{MS[v]} > ect'_u$.

*Proof.* ($\Rightarrow$) *Necessity via contrapositive.* If $LST_{JS[u]} \leq ECT'_u$ or $LST_{MS[v]} \leq ECT'_u$, then by Propositions 2 and 3 the swap cannot reduce the makespan, independently of how many critical paths exist.

($\Leftarrow$) *Sufficiency by contradiction.* Since a decrease in makespan means the LST of the dummy terminal operation $O_T$ decreases, we assume, for contradiction, that $lst'_{O_T} \geq lst_{O_T}$ while $lst_{JS[u]} > ect'_u$ and $lst_{MS[v]} > ect'_u$ both hold. When operations $u$ and $v$ are swapped, all operations on the paths from either $u$ or $v$ to $O_T$ are affected. We call these paths affected paths, excluding the path containing both $u$ and $v$. Since $O_T$ is at the end of all affected paths and $lst_w$ is defined as $\min(lst_{JS[w]}, lst_{MS[w]}) - p_w$, for any operation $w$ on an affected path (except $u$ and $v$), $lst'_w \geq lst_w$ must hold.

The new critical path in the new schedule must include one of the affected paths, since any increase in path length must result from EST adjustments due to the swap. Note that operations before $u$ on the original critical path remain unchanged and must be part of any new critical path. Therefore, we only need to investigate the paths after the swap. All affected paths begin with one of three operation pairs: $(v, JS[v])$, $(u, JS[u])$, or $(u, MS[v])$. We investigate these cases:

(1) Let an affected path starting with $(v, JS[v])$ be the new critical path. We know that $ect_v \leq est_{JS[v]} \leq lst_{JS[v]} \leq lst'_{JS[v]} = est'_{JS[v]} = ect'_v$ by definition of EST and the characteristic of critical path. However, $est'_v < est_v$ must hold. By definition, $est'_v = \max(ect_{JP[v]}, ect_{MP[u]})$ since $ect'_{JP[v]} = ect_{JP[v]}$ and $ect'_{MP[u]} = ect_{MP[u]}$ as these operations are not affected by the swap. Also, since $u$ is on the unique critical path while $JP[v]$ is not, we have $est_v = \max(ect_{JP[v]}, ect_u) = ect_u$ and $ect_{MP[u]} < ect_u$. Therefore, $est'_v = \max(ect_{JP[v]}, ect_{MP[u]}) < ect_u = est_v$. This contradicts $ect_v \leq ect'_v$.

(2) Let an affected path starting with $(u, JS[u])$ be the new critical path. Then $lst_{JS[u]} \leq lst'_{JS[u]} = est'_{JS[u]} = ect'_u$ would hold, contradicting $lst_{JS[u]} > ect'_u$.

(3) Let an affected path starting with $(u, MS[v])$ be the new critical path. Then $lst_{MS[v]} \leq lst'_{MS[v]} = est'_{MS[v]} = ect'_u$ would hold, contradicting $lst_{MS[v]} > ect'_u$.

Since all possible cases lead to contradictions, our assumption $lst'_{O_T} \geq lst_{O_T}$ must be false. Therefore, the makespan decreases when $lst_{JS[u]} > ect'_u$ and $lst_{MS[v]} > ect'_u$.

$\square$

**Theorem 2** For a schedule with multiple critical paths, the makespan strictly decreases when consecutive operations $u$ and $v$ on a machine are swapped *iff*

$$LST_{JS[u]} > ECT'_u, \qquad LST_{MS[v]} > ECT'_u, \qquad \text{and every critical path contains } v.$$

*Proof.* ($\Rightarrow$) *Necessity via contrapositive.* If $LST_{JS[u]} \leq ECT'_u$ or $LST_{MS[v]} \leq ECT'_u$, Propositions 2–3 preclude any decrease. Moreover, if some critical path omits $v$, then by Lemma 1 it can remain unchanged by the swap, preserving the old makespan—contradiction.

($\Leftarrow$) *Sufficiency by contradiction.* Assume the three conditions and suppose $LST'_{O_T} \geq LST_{O_T}$. Every critical path contains $v$, so after the swap any critical suffix must start at one of $(v, JS[v])$, $(u, JS[u])$, $(u, MS[v])$. Using Lemma 2 and the same calculations as in Theorem 1: case $(v, JS[v])$ is impossible because $ECT'_v < ECT_v$; cases $(u, JS[u])$ and $(u, MS[v])$ contradict $LST_{JS[u]} > ECT'_u$ and $LST_{MS[v]} > ECT'_u$, respectively. Thus $LST'_{O_T} < LST_{O_T}$ and the makespan decreases. $\square$

**Theorem 3.** For a schedule with multiple critical paths, if

$$LST_{JS[v]} > ECT'_v, \qquad LST_{JS[u]} > ECT'_u, \qquad LST_{MS[v]} > ECT'_u,$$

then swapping the consecutive pair $(u, v)$ either (i) strictly reduces the makespan or (ii) keeps the makespan but strictly reduces the number of critical paths.

*Proof. Setup.* The three strict inequalities ensure that, after swapping, there is no blocking at $JS[v]$ with respect to $ECT'_v$, nor at $JS[u]$ or $MS[v]$ with respect to $ECT'_u$. Since $p_u > 0$ and $MP[u]$ immediately precedes $u$ on the same machine, we have $ECT_{MP[u]} < ECT_u$; hence

$$EST'_v = \max\{ECT_{JP[v]}, ECT_{MP[u]}\} < \max\{ECT_{JP[v]}, ECT_u\} = EST_v \implies ECT'_v < ECT_v$$

By Lemma 11, only paths intersecting $F^+(u, v)$ may change. By Lemma 2, if the makespan does not decrease ($LST'_{O_T} \geq LST_{O_T}$), then $LST'_w \geq LST_w$ for any $w \in F^+(u, v) \setminus \{u, v\}$.

Consider any original critical path $P$ and classify its relation to $(u, v)$ into six exhaustive, mutually exclusive cases:

(1) **Independent:** $P$ shares no operation with $\{u, v\}$.
(2) **Both:** $P$ contains both $u$ and $v$.

(3) **Share-before-$u$:** $P$ shares some operations before $u$ but excludes $v$.
(4) **Share-all-before-$u$ but exclude $v$:** $P$ shares all operations up to $u$ but excludes $v$.
(5) **Share-after-$v$:** $P$ shares some operations after $v$.
(6) **Share-all-after-$v$ but exclude $u$:** $P$ shares all operations after $v$ but excludes $u$.

**Case (2)** After the swap, any critical suffix must start at one of $(v, JS[v])$, $(u, JS[u])$, or $(u, MS[v])$. However, $(v, JS[v])$ is impossible because $ECT'_v < ECT_v$; starting at $(u, JS[u])$ would force $LST_{JS[u]} \leq ECT'_u$; starting at $(u, MS[v])$ would force $LST_{MS[v]} \leq ECT'_u$. Each contradicts the assumptions. Hence $P$ is no longer critical. If all critical paths are of this type, the makespan strictly decreases; otherwise the number of critical paths strictly decreases.

**Case (5: Share-after-$v$.** Then any critical suffix must start at $(v, JS[v])$, which is impossible because $ECT'_v < ECT_v$. By Lemma 22, $LST'_{JS[v]} \geq LST_{JS[v]} \geq ECT_v > ECT'_v$, so $EST'_{JS[v]} = LST'_{JS[v]} = ECT'_v$ cannot hold. Thus $P$ is no longer critical; the same conclusion as in Case (2) follows.

**Case (3)** If $P$ remains critical, its critical suffix must start at $(u, JS[u])$, which requires $LST'_{JS[u]} = EST'_{JS[u]} = ECT'_u$. By Lemma 2, $LST'_{JS[u]} \geq LST_{JS[u]}$, hence $LST_{JS[u]} \leq ECT'_u$, contradicting $LST_{JS[u]} > ECT'_u$. Thus $P$ is not critical; the same conclusion as above holds.

**Cases (4) and (6)** For Case (4), a critical suffix from $(u, JS[u])$ would enforce $LST_{JS[u]} \leq ECT'_u$, contradicting $LST_{JS[u]} > ECT'_u$. For Case (6), a critical suffix via $(u, MS[v])$ would enforce $LST_{MS[v]} \leq ECT'_u$, contradicting $LST_{MS[v]} > ECT'_u$.

**Case (1)** Such paths can keep their length (Lemma 1). However, by the strict advance at $v$ and the absence of blocking at $JS[v]$, $JS[u]$, and $MS[v]$, at least one of the other cases occurs and drops from criticality.

Therefore either the makespan decreases (if all critical paths fall) or the number of critical paths strictly decreases.

$\square$

# E    ADAPTIVE REVISITING CRITERIA

The length of list that saves recently swapped operation pairs is randomly selected between the given minimal and maximal values $L_{\min}$ and $L_{\max}$ simply for each instance, following the adaptive tabu strategy proposed by Zhang et al. (2007). The $L_{\min}$ and $L_{\max}$ are computed as follows:

$$L = 10 + \frac{N}{M}$$

$$L_{\min} = \left\lfloor L + \frac{1}{2} \right\rfloor$$

$$L_{\max} = \begin{cases} \left\lfloor 1.4L + \frac{1}{2} \right\rfloor, & \text{if } N \leq 2M \\ \left\lfloor 1.5L + \frac{1}{2} \right\rfloor, & \text{otherwise} \end{cases}$$

# F    DETAILED LEARNING PROCESS

Training loop including trajectory collection, gradient computation, and parameter updates is described in Algorithm 1.

# G    CONFIGURATIONS OF EXPERIMENTS

Configurations including activation functions, hyperparameters, and hardware settings are shown in Table 3.

---

**Algorithm 1** Entropy Regularized $n$-step REINFORCE

---

**Input**: training problem size $N \times M$, validation instances $\mathcal{I}^{val}$
**Parameter**: batch size $B$, # of epochs $N^{epoch}$, # of steps per epoch $T$, learning period $d^{learn}$, validation period $d^{val}$, learning rate $\alpha$, strength of entropy regularization $\beta$
**Output**: best parameter set $\theta^{best}$

1: Initialize $\theta$, $\theta^{best} = \theta$, $\bar{\mathcal{C}}^{best} = \infty$
2: **for** $epoch = 1$ to $N^{epoch}$ **do**
3:     Generate $B$ instances with $N$ jobs and $M$ machines
4:     Initialize schedules $\{s_0^1, ..., s_0^B\}$ by using FDD/MWKR rule
5:     **for** $t = 0$ to $T$ **do**
6:         **for** $s_t^b \in s_0^1, ..., s_0^B$ **do**
7:             Sample an action $a_t^b \sim \pi_\theta(a_t^b | s_t^b)$
8:             Derive $s_{t+1}^b$, $r(a_t^b, s_t^b)$, and $\mathcal{H}(\pi_\theta(\cdot | s_t^b))$ by $a_t^b$
9:         **end for**
10:        **if** $t$ mod $d^{learn} = 0$ **then**
11:           Compute $\bar{R}$ by normalizing cumulative rewards
12:           $\mathcal{L}(\theta) = -\sum_{b=1}^{B}\sum_{j=0}^{d^{learn}}[\bar{R}_{t-j}^b \log \pi_\theta(a_{t-j}^b | s_{t-j}^b) + \beta\mathcal{H}(\pi_\theta(\cdot | s_{t-j}^b))]$
13:           $\theta \leftarrow \text{Adam}(\theta, \nabla_\theta \mathcal{L}(\theta))$
14:        **end if**
15:     **end for**
16:     **if** $epoch$ mod $d^{val} = 0$ **then**
17:        $\bar{\mathcal{C}}$ = mean of objectives for $\mathcal{I}^{val}$ with $\pi_\theta$
18:        **if** $\bar{\mathcal{C}} < \bar{\mathcal{C}}^{best}$ **then**
19:           $\bar{\mathcal{C}}^{best} = \bar{\mathcal{C}}$, $\theta^{best} = \theta$
20:        **end if**
21:     **end if**
22: **end for**

---

Table 3: Model and Training Configuration.

| Component | Setting |
|---|---|
| Encoder activation function | LeakyReLU |
| Decoder activation function | tanh |
| Optimizer | Adam |
| MLP architecture | 4 layers, 512 hidden units |
| Encoder output dimension | 128 |
| Batch size ($B$) | 64 |
| Epochs ($N^{epoch}$) | 2000 |
| Steps per epoch ($T$) | 500 |
| Learning period ($d^{learn}$) | 10 |
| Validation period ($d^{val}$) | 10 |
| Learning rate ($\alpha$) | 1e–5 |
| Entropy regularization strength ($\beta$) | 1e–5 |
| CPU | Intel Core i7-7700K @ 4.20GHz |
| GPU | NVIDIA GeForce RTX 4090 |

## H ACTION SELECTION ANALYSIS

## I ABLATION STUDIES

Table 5 shows the results of five different ablation studies: the effectiveness of (1) condition indicators with different encoders, (2) different types of proposition identifiers, (3) different state features, (4) different neighborhood structures, and (5) different sizes of training instances. In the table, 'Gap' represents the average optimality gap for 162 JSSP instances, and 'Diff' represents the average difference in optimality gap from the best-performing method of each instance in each block. A 'Diff' of 0% indicates that the method consistently achieved the best performance across all instances.

Table 4: Satisfaction ratio (%) of proposition conditions and tabu across iterations and JSSP instance groups.

| Condition | # of iterations | JSSP Instance Group | | | | |
|---|---|---|---|---|---|---|
| | | TA 15×15 | TA 20×15 | TA 20×20 | TA 30×15 | TA 30×20 |
| including tabu list | 500 | 23.3 | 22.5 | 14.9 | 25.6 | 19.2 |
| | 1000 | 25.1 | 27.8 | 16.2 | 29.4 | 22.5 |
| | 5000 | 24.6 | 33.1 | 21.9 | 39.2 | 27.4 |
| $lst_{JS[v]} > ect'_v.$ | 500 | 99.8 | 99.8 | 99.6 | 99.7 | 99.7 |
| | 1000 | 99.8 | 99.8 | 99.6 | 99.7 | 99.7 |
| | 5000 | 99.7 | 99.8 | 99.6 | 99.7 | 99.7 |
| $lst_{JS[u]} > ect'_u$ | 500 | 91.0 | 89.5 | 91.1 | 87.3 | 90.0 |
| | 1000 | 90.4 | 90.0 | 91.1 | 87.3 | 90.0 |
| | 5000 | 90.9 | 87.7 | 89.6 | 88.9 | 88.6 |
| $lst_{MS[v]} > ect'_u$ | 500 | 84.6 | 85.7 | 89.0 | 92.9 | 88.0 |
| | 1000 | 83.9 | 82.9 | 88.1 | 91.3 | 87.4 |
| | 5000 | 84.0 | 83.8 | 87.5 | 89.3 | 85.8 |

Table 5: Results of Ablation Studies.

| Method | # of iterations | | | | | |
|---|---|---|---|---|---|---|
| | 500 | | 1000 | | 5000 | |
| | Gap | Diff | Gap | Diff | Gap | Diff |
| None + MLP | 12.5% | 4.3% | 11.1% | 4.2% | 8.1% | 3.8% |
| None + bi-GAT (TBGAT) | 10.6% | 2.4% | 9.3% | 2.5% | 6.9% | 2.6% |
| indicators + MLP (LSI) | 8.9% | 0.7% | **7.5%** | **0.6%** | **4.9%** | **0.6%** |
| indicators + bi-GAT | **8.9%** | **0.7%** | 7.5% | 0.6% | 5.1% | 0.8% |
| intersection indicator | 9.1% | 0.8% | 7.7% | 0.8% | 5.1% | 0.7% |
| indicators (LSI) | **8.9%** | **0.6%** | **7.5%** | **0.6%** | **4.9%** | **0.6%** |
| values | 10.1% | 1.7% | 9.0% | 2.1% | 6.9% | 2.6% |
| normalized values | 9.8% | 1.5% | 8.8% | 2.0% | 6.5% | 2.2% |
| simple (LSI) | 8.9% | 1.1% | 7.5% | 1.0% | 4.9% | 0.9% |
| simple + topological order | **8.6%** | **0.7%** | **7.2%** | **0.8%** | 4.8% | 0.8% |
| simple + instance-dependent | 8.8% | 1.0% | 7.4% | 1.0% | **4.8%** | **0.8%** |
| $N1$ multiple | 9.5% | 1.5% | 8.1% | 1.5% | 5.4% | 1.2% |
| $N1$ | 9.6% | 1.6% | 8.3% | 1.7% | 5.4% | 1.3% |
| $N5$ multiple (LSI) | 8.9% | 0.9% | **7.5%** | **0.8%** | 4.9% | 0.8% |
| $N5$ | **8.9%** | **0.9%** | 7.5% | 0.9% | **4.9%** | **0.8%** |
| $N5$ + action masking | 17.6% | 9.5% | 17.6% | 10.9% | 17.6% | 13.4% |
| LSI 10x10 | **8.9%** | **1.0%** | **7.5%** | **0.7%** | **4.9%** | **0.6%** |
| LSI 15x15 | 9.3% | 1.4% | 8.2% | 1.4% | 5.8% | 1.5% |
| LSI 20x20 | 9.6% | 1.7% | 8.5% | 1.7% | 6.0% | 1.7% |
| LSI 30x20 | 9.4% | 1.5% | 8.2% | 1.4% | 6.2% | 1.9% |
| TBGAT 10x10 | 10.6% | 2.7% | 9.3% | 2.6% | 6.9% | 2.6% |
| TBGAT 15x15 | 10.7% | 2.8% | 9.6% | 2.9% | 7.3% | 3.0% |
| TBGAT 20x20 | 11.0% | 3.1% | 10.2% | 3.5% | 8.7% | 4.4% |
| TBGAT 30x20 | 10.2% | 2.3% | 9.1% | 2.4% | 6.9% | 2.7% |

The inclusion of condition indicators significantly improves performance for both encoder types, a simple MLP and TBGAT's bidirectional topological GAT (bi-GAT). Both encoders with indicators achieve almost identical performance, differing by less than 0.2%. This suggests that our theoretically-derived indicators can effectively replace complex neural structures designed to learn problem characteristics.

For indicator types, we compared our approach with three alternatives: (1) intersection indicator, which denotes satisfying all three propositions' conditions simultaneously, (2) values, which represent the differences between the left and right sides of propositions' conditions, and (3) normalized values,

where each difference is normalized by the maximal processing time of operations. Both our approach and intersection indicator outperform value-based approaches, showing nearly identical performance, differing by less than 0.2%.

The impact of state features was examined by comparing variants with additional features: the topological order of operations used in TBGAT and instance-dependent features, including features used in SN or IRD. In contrast, LSI uses three simple state features: processing time, EST, and LST of operations. These additional features provided only minor improvements with 0.1–0.3% lower Gap, suggesting that simple features are sufficient.

For neighborhood structures, while $N1$ theoretically guarantees optimal solution reachability, $N5$ showed better empirical performance with 0.5–0.8% lower Gap, probably due to its more focused search space. Considering multiple critical paths ($N5$ multiple) performed similarly to considering a single critical path randomly chosen ($N5$). However, using intersection indicator for action masking led to premature convergence to local optima, resulting in worse performance.

Training with instances of different sizes (10x10 to 30x20) showed that larger training instances did not necessarily lead to better performance. Interestingly, training LSI with the smallest instances (10x10) demonstrates the best performance.

## J  EXTENDED EXPERIMENTAL ANALYSIS

We conducted an ablation study to assess the individual effects of the components of our approach. We first test for the effectiveness of condition indicators and the encoder's structure. The results are shown in Figures 9. In these figures, the x-axis represents benchmark groups, while the y-axis shows the optimality gap. Results are shown with different iteration numbers. The performance remains consistent whether using a simple MLP or TBGAT's bidirectional topological GAT (bi-GAT) as an encoder structure when condition indicators are incorporated. Without condition indicators, the bi-GAT structure performs better than MLP, yet still underperforms compared to approaches using condition indicators.

We further examined various indicator types, with the results presented in Figures 10. The 'label_v' represents the numerical difference between the left and right-hand sides of the propositions' conditions, while 'label_v_norm' denotes this value normalized by maximal processing time. 'label_intersection' sets the indicator to 1 only when the conditions of all three propositions are simultaneously satisfied, while 'label_l' denotes our proposed approach that considers the conditions individually. Although the performance difference between 'label_intersection' and individual condition consideration was not substantial, considering conditions independently generally showed slightly better performance.

TBGAT utilized not only processing time, EST, and LST but also topological order of each operation as operation features. We investigated the effectiveness of incorporating this topological information and the operation features used in the dispatcher from Section 4.3. As illustrated in Figures 11, the inclusion of these additional features demonstrated negligible impact on performance enhancement.

We also investigated the impact of neighborhood structures used in generating candidate moves. Finally, we examined the effectiveness of different neighborhood structures in generating candidate moves. While N5 neighborhood structure contains all makespan-improving moves, it lacks the theoretical guarantee of optimal solution reachability that N1 neighborhood structure possesses with its broader action space. However, as shown in Figures 12, using N5 neighborhood structure experimentally outperformed using N1 neighborhood structure. Furthermore, considering multiple critical paths simultaneously with N5 neighborhood structure ('N5_multi') showed slightly better performance than randomly selecting a single critical path ('N5'). Additionally, the case where condition indicators from the decoder were used for action masking is denoted as 'N5_improve' in the figure. This approach appears to have converged prematurely to local optima before reaching 500 iterations.

Finally, we test for different training problem sizes. We conducted experiments with increasing problem sizes from 10x10 to 20x20, and the results are shown in Figures 13, which include the ranges of optimal gaps across three replications for our proposed approach. Counter to intuition, we observed performance degradation in some JSSP instances even when the training problem size was

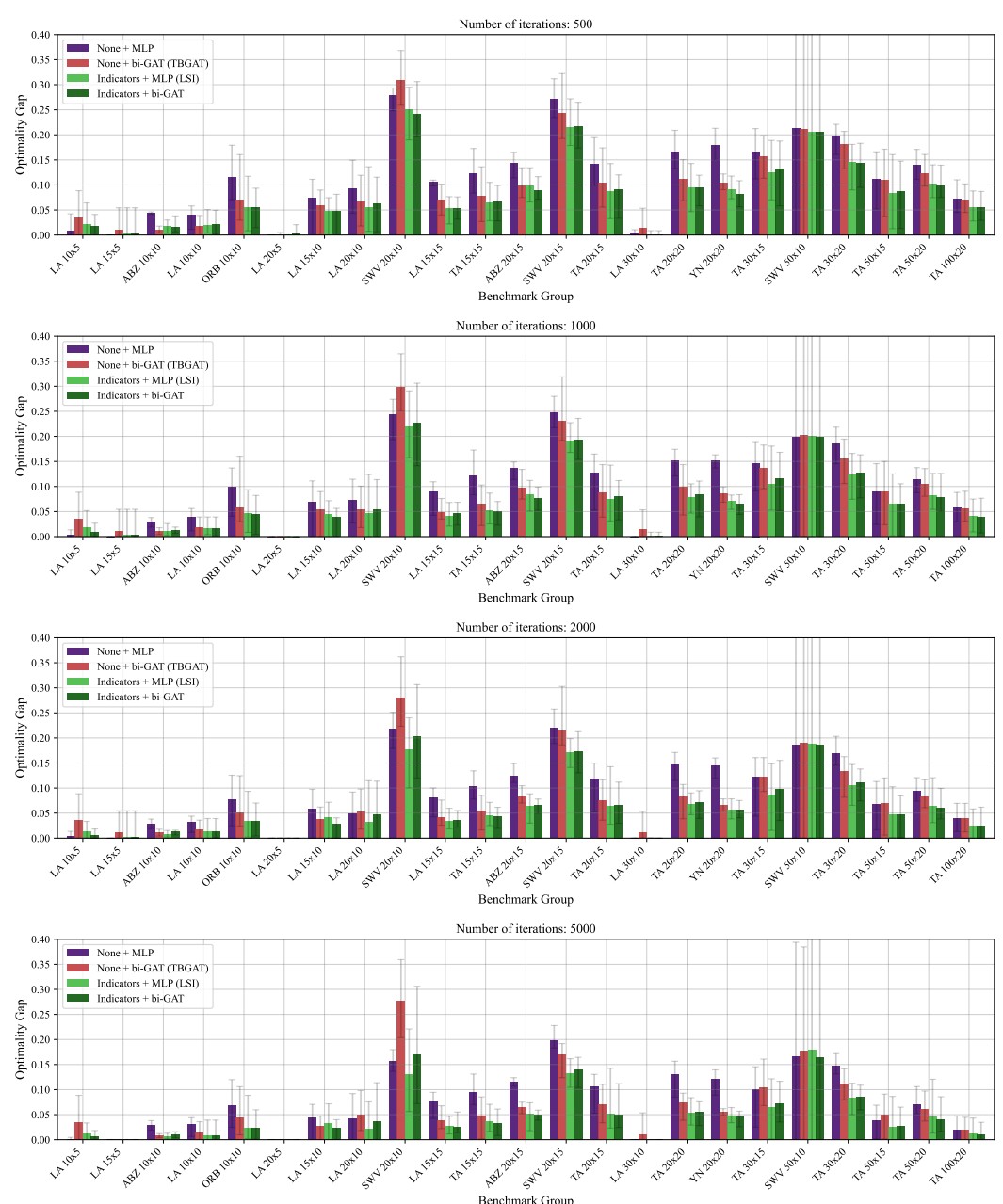

Figure 9: Mean optimality gap for benchmarks groups with different policy network structures and different iteration numbers.

closer to the size of the target instances. This suggests that broader solution spaces in the training process might hinder convergence to effective policies.

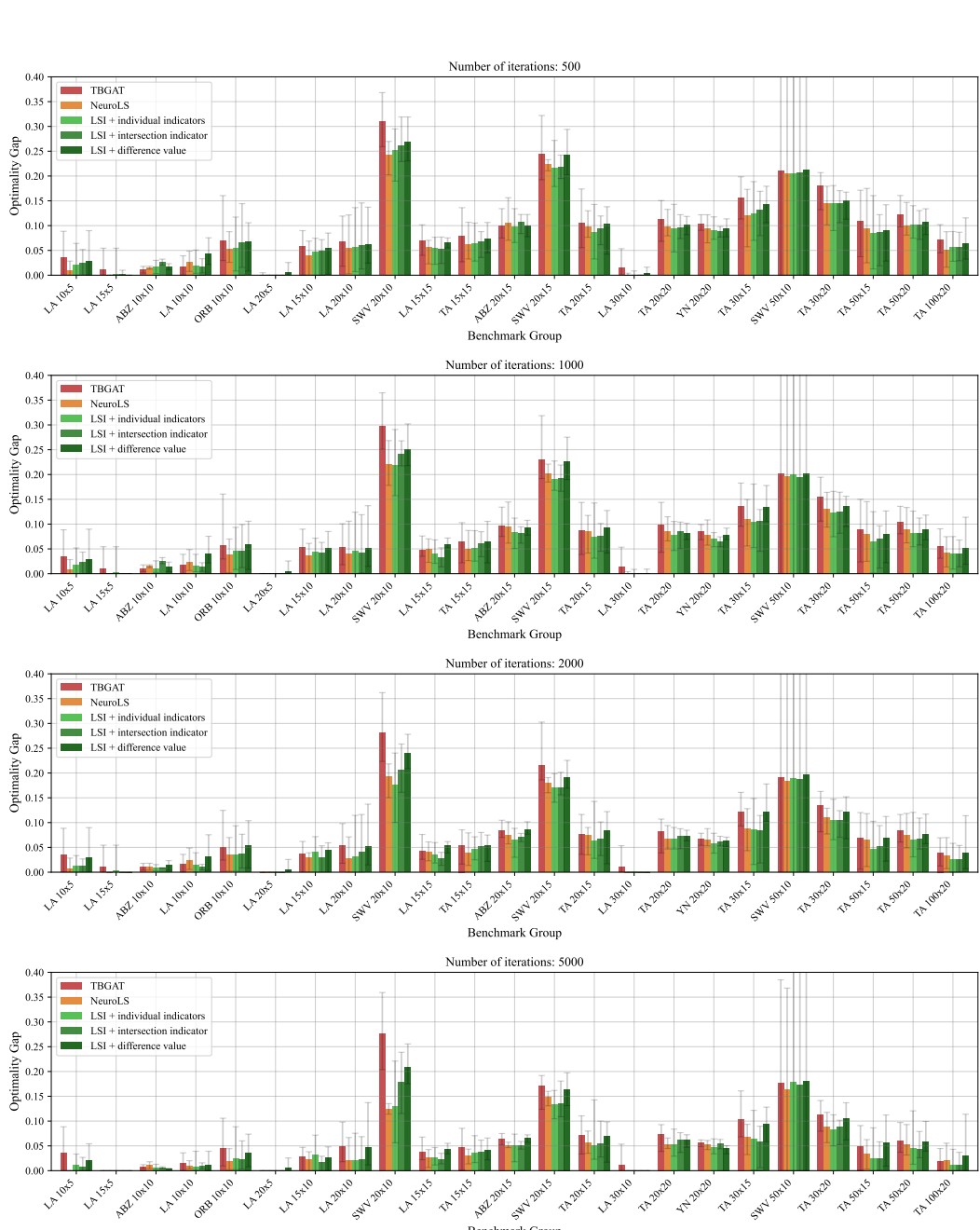

Figure 10: Mean optimality gap for benchmarks groups with different ways to corporate the theoretically derived conditions for makespan reduction and different iteration numbers.

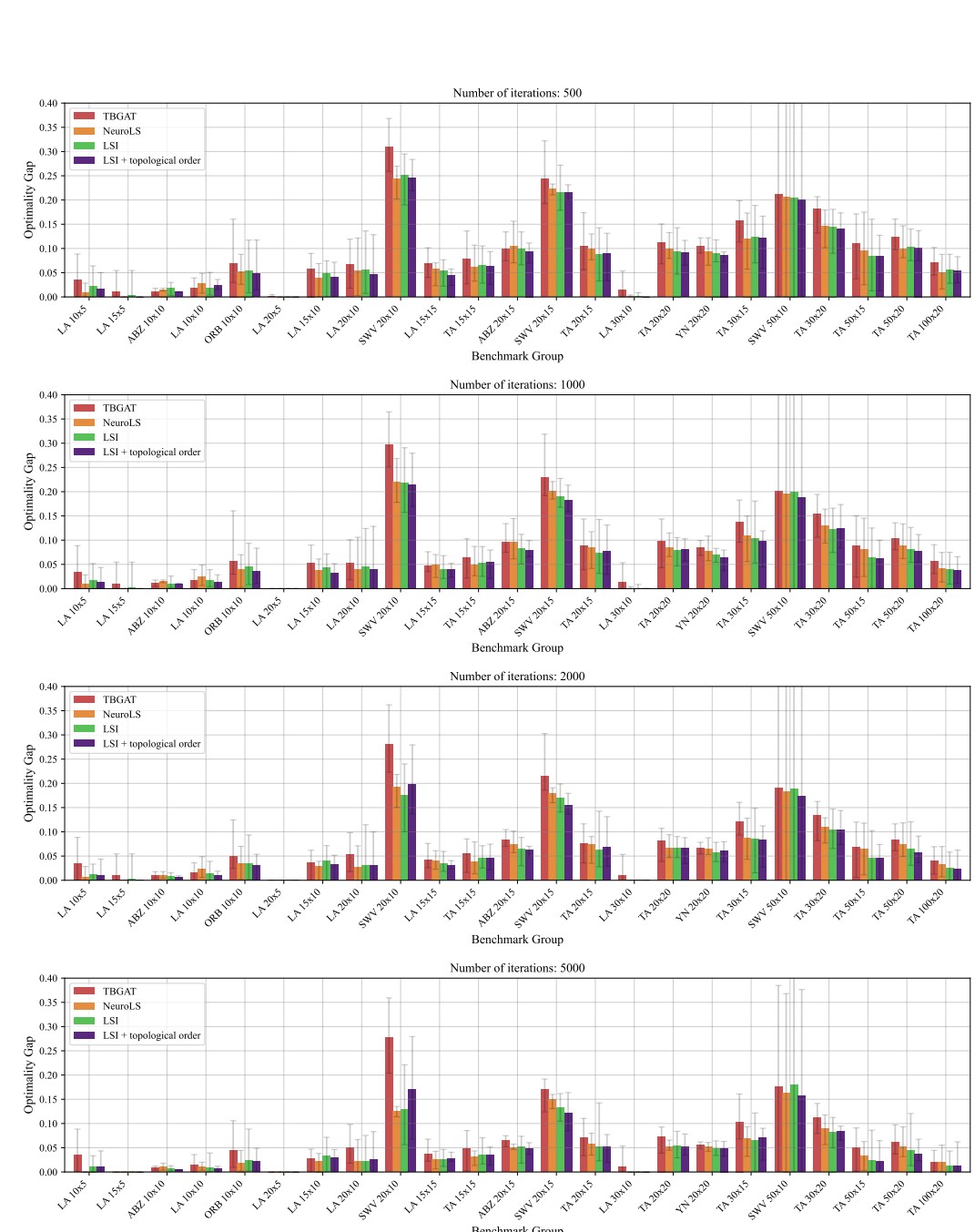

Figure 11: Mean optimality gap for benchmarks groups with different input features and different iteration numbers.

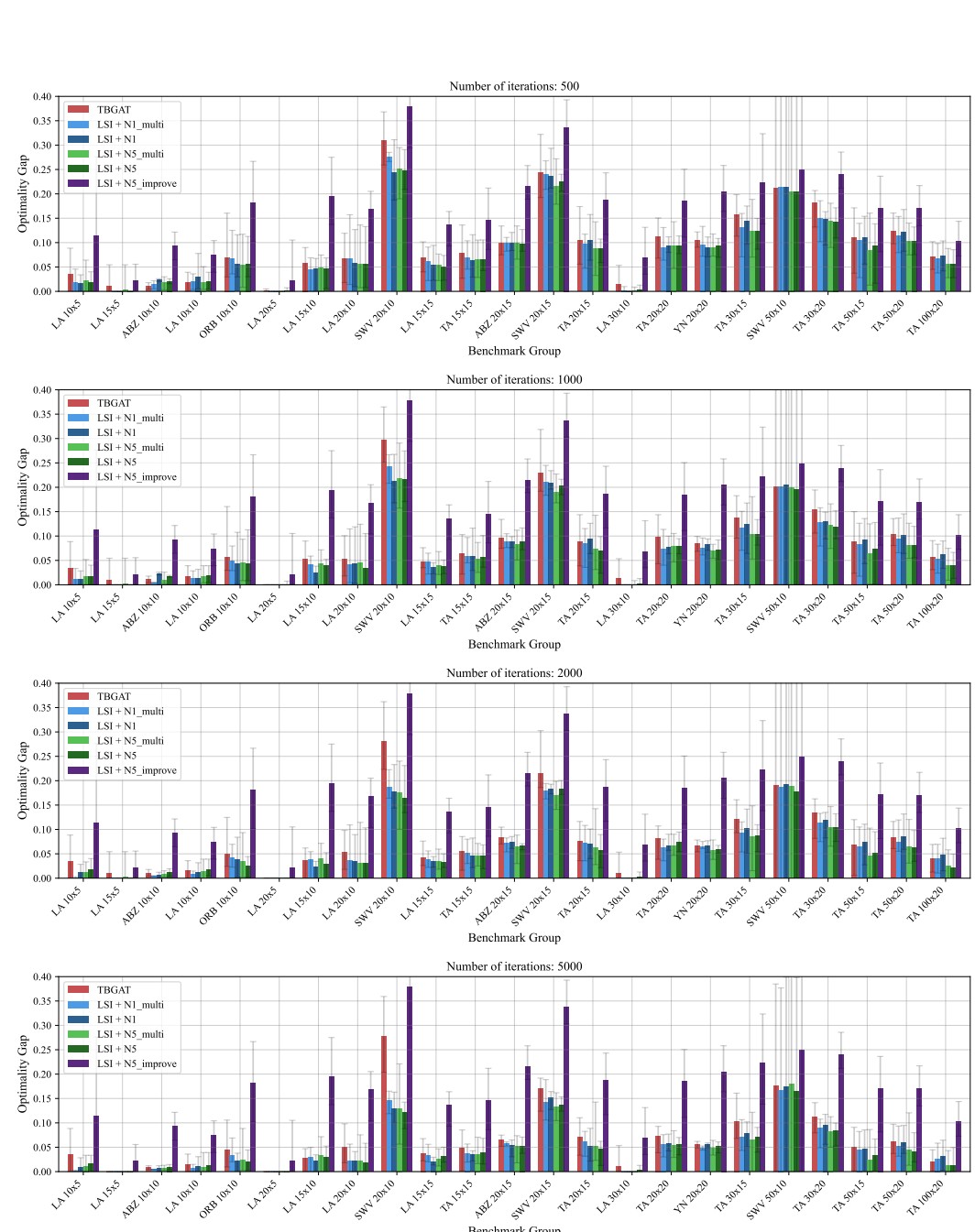

Figure 12: Mean optimality gap for benchmarks groups with different neighborhood structures and different iteration numbers.

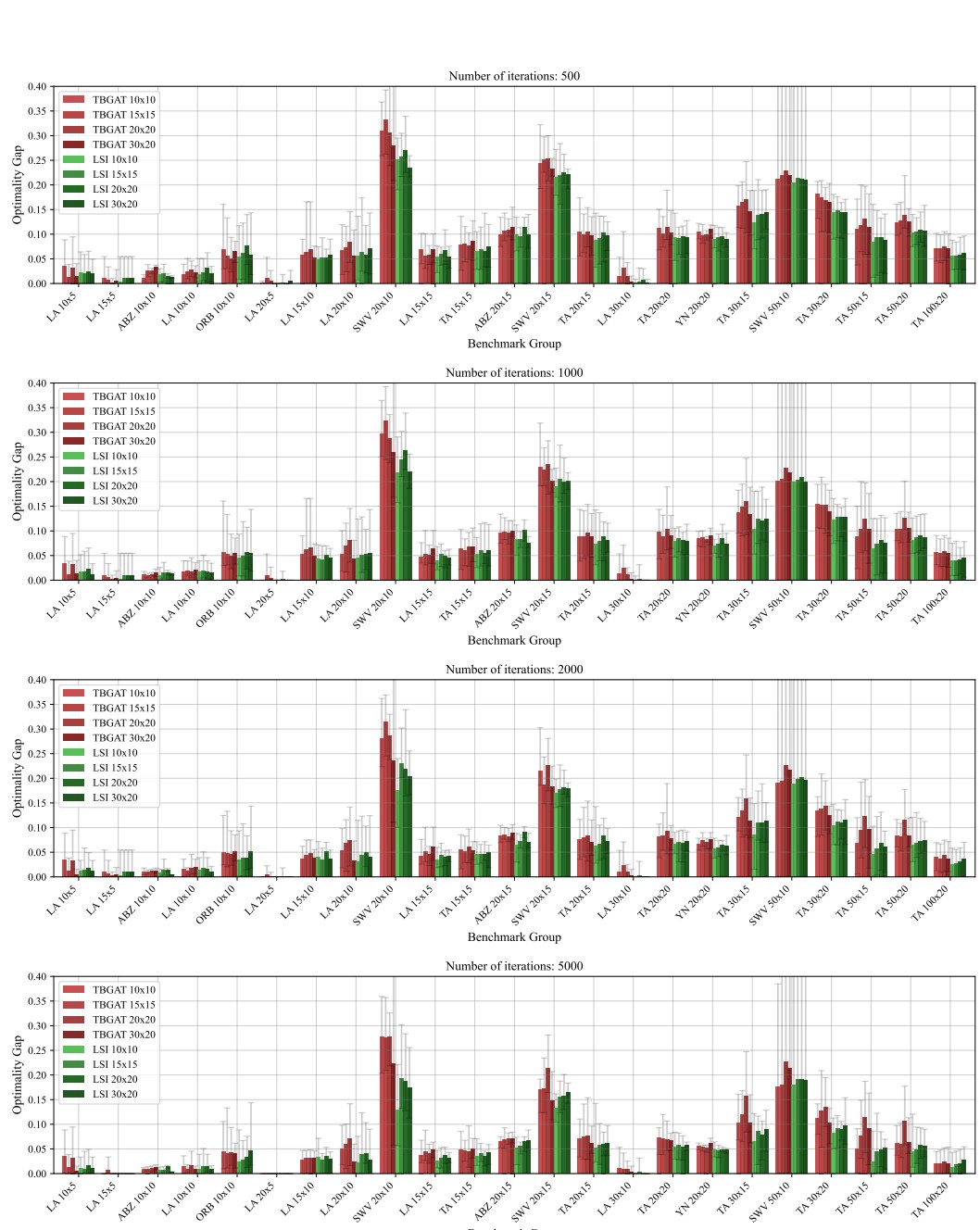

Figure 13: Mean optimality gap for benchmarks groups with different training instance sizes and different iteration numbers.

