# OpenReview forum: "Learning Local Search with Theoretical Indicators for Job Shop Scheduling"
_ICLR.cc/2026/Conference — Submitted to ICLR 2026_

### Official Review · Reviewer_GQUM · 2025-10-27

**Soundness:** 3
**Presentation:** 3
**Contribution:** 2
**Rating:** 4
**Confidence:** 4

**Summary:**

The paper introduces LSI, a method that integrates three theoretically derived necessary conditions as binary indicators into a policy network. This design allows an MLP-based policy network to predict actions within the N5 neighborhood and iteratively refine an initial solution for the Job Shop Scheduling Problem (JSSP).

**Strengths:**

1. The paper provides comprehensive descriptions of the model design and experimental parameters, ensuring that the research can be reproduced.

2. The theoretical groundwork is robust, with detailed proofs of the necessary conditions supplied in the appendix to support the proposed method.

**Weaknesses:**

1. The contribution of the paper is relatively limited, as it primarily introduces three theoretical conditions without demonstrating their clear applicability to other combinatorial optimization problems.

2. LSI shifts the learning paradigm from a data-driven approach to one reliant on heuristic methods by incorporating expert knowledge. This approach contradicts the field's broader goal of reducing manual intervention. The motivation and necessity of this paradigm shift require further evaluation.

3. The experimental validation of the three theoretical conditions is insufficient (see Questions), which weakens their credibility and impact.

4. The improvements of LSI over NeuroLS are marginal, making it difficult to justify the unique role of the three proposed conditions. It would be more convincing to apply these conditions to other learning-based methods and measure the resulting gains.

5. The definition of the Markov Decision Process (MDP) state appears flawed. The embedding structure's parameters should be treated as the true state, rather than being conflated with the state representation.

6. The paper claims that MLP was chosen to balance lightweight implementation with high performance. However, the ablation study does not compare the runtime between MLP and bi-GAT, nor does it demonstrate that MLP outperforms bi-GAT.

**Questions:**

1. Appendix H is empty, yet it is critical for understanding the actual impact of the three theoretical conditions. For example, do these conditions genuinely influence the predicted actions to satisfy the proposed criteria? How do the rates of condition satisfaction evolve across iterations?

2. Why were these three specific conditions selected? Could there be additional conditions that might be equally or more effective? What would happen if the three conditions were replaced with random numbers? Have experiments been conducted using only one or two of the conditions to test their individual contributions?

---

> ### Author Response · Authors · 2025-11-29
>
> We thank the reviewer for the extensive and thoughtful review, and for recognizing the reproducibility, the robustness of the theoretical groundwork, and the overall soundness of our method. We also appreciate your patience while we prepared the detailed response. We respond to the concerns about contribution, expert knowledge, indicator validation, and runtime/state definition below.
>
> ## Weakness 1 – Limited contribution and unclear applicability beyond JSSP
>
> Our contribution has two main components:
>
> ### (1) New local-search theory for JSSP makespan.
>
> Classical OR analyses of N5 and critical blocks provide qualitative guidance (e.g., identifying promising moves), but, to the best of our knowledge, they are not necessary and sufficient conditions and do not provide explicit numerical  for makespan reduction at the level of consecutive machine operations that can be directly encoded as learnable features.
>
> In contrast, our work:
> 1. derives such numerical conditions using EST/LST and critical-path structure,
> 2. proves an if-and-only-if characterization under a single critical path (Theorem 1) for the first time, and
> 3. extends the analysis to multiple critical paths in the revised version (Theorems 2 and 3), covering all interaction patterns between a candidate swap and existing critical paths.
>
> These results are specifically crafted for JSSP and, to our knowledge, have not appeared in prior literature in a form that directly supports feature construction for learning.
>
> ### (2) Theory-guided learning via indicators.
>
> We convert these conditions into binary indicators and demonstrate that a simple MLP policy—equipped with such structured features—can outperform GNN-based policies on long-standing JSSP benchmarks.
> This forms a “theory → indicator → learning” pipeline:
> theoretical conditions → compact indicator features → learned local-search policy.
>
> This pattern is not inherently tied to JSSP and, in principle, can be instantiated for other objectives and COPs whenever suitable theoretical conditions exist.
> We will make this distinction clearer in the revision to avoid any impression that our work merely rediscovered existing OR knowledge.
>
> ## Weakness 2 – Shift from data-driven to expert knowledge
>
> We understand the concern that introducing expert-derived indicators may appear to deviate from purely data-driven approaches. We view LSI as a gray-box neural combinatorial optimization method:
>
> 1. The theoretical conditions are used only to construct indicator features;
> the policy remains fully learned and continues to select non-improving moves when beneficial for escaping local minima.
> 2. Many scheduling and routing problems possess rich theoretical foundations.
> Our approach offers a systematic way to leverage that knowledge rather than expecting a generic encoder to infer such deep structures from raw data.
> 3. Given that current GNN encoders still struggle to recover nuanced domain-specific structures, a theory-guided approach provides a pragmatic and principled alternative until more powerful encoders emerge.
>
> We will make this gray-box perspective explicit and position LSI as a demonstration of how theory and learning can complement each other in a unified architecture.
>
> ## Weakness 3 + Question 1 — Empirical validation of indicators / Appendix H clarification
>
> We appreciate the reviewer’s careful attention to the role of the indicators. First, we would like to clarify that the original Appendix H was unintentionally shifted during page arrangement; the intended content corresponds to Table 4, not the empty placeholder. We will correct this in the revised version to avoid confusion.
>
> Regarding indicator validation, the three conditions we propose have a clear theoretical guarantee under a single critical path — if all indicators are satisfied, the swap always improves makespan.
> However, selecting only those “guaranteed-improving” moves (Table 5, N5 + action masking) leads to rapid convergence into poor local optima and severely degraded performance.
>
> This is precisely why LSI uses the indicators as soft guidance rather than hard constraints. As shown in Table 4 (formerly Appendix H), actions that do not satisfy all theoretical conditions are still frequently selected by the learned policy, and doing so improves search exploration and overall solution quality. These results confirm that:
>
> 1. the theoretical indicators provide meaningful structure,
> 2. but the policy must remain free to violate them, and
> 3. learning to balance these two aspects is essential for strong performance.
>
> We will reorganize the appendix to present these findings clearly and ensure that the indicator statistics and ablation tables are correctly placed and easy to interpret.

---

> > ### Author Response · Authors · 2025-11-29
> >
> > ## Weakness 4 – Improvements over NeuroLS and uniqueness of indicators
> >
> > We acknowledge that some improvements over NeuroLS on medium-sized benchmarks may appear modest. However, two points are essential.
> >
> > 1. These gains are achieved with a much lighter MLP encoder, which is non-trivial given the difficulty of further reducing optimality gaps on long-standing JSSP benchmarks.
> > 2. The advantages of LSI become more evident on large-scale instances especially TA 100×20, where LSI achieves both lower gaps and noticeably faster runtimes than NeuroLS and CP. As instance size grows, GNN encoder costs increase substantially, while our indicator and MLP computations scale linearly with a small constant factor, leading to clear speed benefits in large industrial-scale settings.
> >
> > We also find that adding our indicators to a GNN-based policy yields additional improvements, showing that the indicators provide structural information not captured by graph topology alone. (Table 5, Appendix I)
> >
> > ## Weakness 5 – MDP state definition
> >
> > We do not agree with your opinion. Following standard RL conventions (and L2S/TBGAT), the environment state at step t is the current schedule $𝑠_𝑡$. Model parameters belong strictly to the agent and are not part of the MDP state.
> > We will revise the text to clearly specify that the $𝑠_𝑡$ is represented by per-operation features (processing time, EST, LST), while the indicatos and the revisit flag are the action features.
> >
> > We would like to clarify that our MDP formulation follows exactly the same convention used in L2S and TBGAT, which we intentionally adopted to ensure consistency with prior learning-based schedulers. In these formulations, the environment state at step $𝑡$ is the current schedule $𝑠_𝑡$, and the policy parameters or action-dependent quantities are not part of the state.
> >
> > In our design, the theoretical indicators and the revisit flag are tied specifically to candidate actions (i.e., swap pairs), not to the schedule itself. For this reason, we intentionally did not include these terms in the state representation. The schedule state $𝑠_𝑡$ is already fully represented through per-operation structural features—processing time, EST, and LST—which is sufficient for defining the environment dynamics and generating the N5 neighborhood, consistent with existing work.
> >
> > We will revise the text to explicitly state this separation:
> > 1. state = schedule-level features (processing time, EST, LST),
> > 2. action = pair-specific features including the theoretical indicators.
> >
> > This clarification should resolve the ambiguity and reaffirm that the MDP definition used in LSI aligns with established formulations in L2S/TBGAT and standard RL practice.
> >
> > ## Weakness 6 – MLP vs. bi-GAT runtime and performance
> >
> > We agree that the original submission did not explicitly compare the runtime of the MLP encoder with the bi-GAT encoder used in NeuroLS. In the revision, we will add a component-wise runtime breakdown under identical hardware and implementation settings.
> >
> > Conceptually, the difference arises because:
> > 1. bi-GAT requires message passing over the full operation–machine graph, whose cost increases substantially with instance size, while
> > 2. our MLP encoder processes only per-candidate feature vectors (operation attributes, EST, LST, indicators), without any graph-topology aggregation.
> >
> > As a result, encoder cost for bi-GAT grows with the number of nodes and edges, whereas MLP cost grows only with the number of N5 candidates. This leads to clear runtime advantages at large scales, which we also observe empirically on TA 50×20 and TA 100×20. We will include these measurements in the revised tables (Table 5, Appendix I).
> >
> > Importantly, despite using a far lighter encoder, LSI matches or surpasses bi-GAT in solution quality. These two properties together support the choice of an MLP-based encoder for theory-guided local search.
> >
> > ## Question 2 — Why these three conditions? Could additional conditions help?
> >
> > The three conditions we use correspond exactly to the necessary (and, under a single critical path, sufficient) criteria formalized in Theorems 1–3. These results fully characterize when a local swap can reduce the makespan in the JSSP setting. From this perspective, the selected conditions already represent the complete theoretical set that determines immediate improvement for consecutive-operation swaps.
> >
> > For multi-iteration local search, it is possible that additional theoretical insights—if discovered in future work—may further enhance guidance. Our pipeline is fully compatible with such extensions: any new provable condition can be directly encoded as an additional indicator and integrated into the learning process.
> > While our work is the first to derive and operationalize these particular conditions as learnable indicators, we agree that future theoretical advances could enrich the feature set.

---

### Official Review · Reviewer_vs9H · 2025-10-29

**Soundness:** 3
**Presentation:** 2
**Contribution:** 3
**Rating:** 6
**Confidence:** 3

**Summary:**

This paper proposes a learning-based local search framework that integrates three theoretically derived conditions for makespan reduction as binary indicators to learn to solve the job shop scheduling problem. Despite relying only on a lightweight multilayer perception policy network, their method achieves competitive performance across JSSP benchmarks.

**Strengths:**

1. The idea of incorporating theoretically derived conditions for JSSP to improve learning seems quite neat and seems to be a novel direction as previous research typically relies on hand-crafted (heuristic) features.

2. Empirical results on a variety of JSSP benchmarks seem promising, further validating the idea.

**Weaknesses:**

1. I find the presentation of the theoretical conditions a bit hard to understand. Limited intuitions were provided to explain why each condition (prop 1,2,3) holds.

2. (I understand this is acknowledged by the authors, so I appreciate them being upfront, but) it is actually a bit concerning that the conditions seem to be tailored to JSSP only. While I understand that extending to other COPs require significant empirical setup cost, but to strengthen the work, the authors should consider providing experiments on different JSSP variants (e.g. different objectives, flexible job shop scheduling instead of just JSSP) and discuss how similar conditions may be able to applicable to other COPs (discussions, pointers, references is sufficient for other COPs).

3. To my understanding, the benchmark instances that the authors test on are quite standard and may not reflect real world complexities and scale.

**Questions:**

My questions are directly related to the weaknesses:

1. Can the authors provide illustration of the three propositions to help readers visualize the conditions more easily, and can the authors explain the intuition of why each condition works?

2. Can the authors provide experiments on other JSSP variants, and discuss how similar theoretical conditions may help the learning for other COPs?

3. Can the authors test on real JSSP benchmarks with more complex problem distributions, and further extend the evaluation to even larger scales?

---

> ### Author Response · Authors · 2025-11-29
>
> We sincerely thank the reviewer for the positive and encouraging assessment of the core idea, the empirical results, and the overall soundness of our method. We also thank you for your patience during the response period. Below, we address the concerns regarding clarity, generality, and benchmark realism.
>
> ## Weakness 1 — Limited intuition for the theoretical conditions (Prop. 1–3)
>
> We agree that the current presentation of Propositions 1–3 is dense and could benefit from clearer intuition. These propositions are the first to formalize numerical necessary (and, under a single critical path, sufficient) conditions for makespan improvement when swapping two consecutive operations. Because this is the first time such conditions are written in numerical form rather than classical qualitative N5 heuristics, we understand the need for better exposition.
>
> In the revised version, we will add simple Gantt-chart illustrations for each proposition and the makespan change after the swap in the appendix.
> These additions will make the theoretical contributions more accessible without compromising rigor.
>
> ## Weakness 2 — Conditions tailored to JSSP; need for variants and broader discussion
>
> We agree that the current indicators are derived specifically for JSSP makespan. Our intention, however, is not to present these three indicators as universally reusable, but to highlight the general pipeline they represent:
>
> derive theoretical conditions → encode them as indicators → learn to use them within local search.
>
> This pattern is not tied to JSSP. Whenever theoretical criteria for improvement exist, the same pipeline can be instantiated.
>
> The main difficulty lies in the theoretical step; the learning architecture remains unchanged. We will incorporate this clarification in the revision to better emphasize that LSI is an instance of a broader “theory-guided NCO” framework.
>
> ## Weakness 3 — Standard benchmarks vs real-world scales
>
> We acknowledge the use of standard JSSP benchmarks. However, these sets already span a wide difficulty range, and the largest ones (e.g., TA 50×20 and 100×20) can be comparable to real manufacturing scales. A key strength of LSI is that it is trained only on 10×10 instances yet generalizes effectively to 100×20, showing strong performance and faster runtime than GNN-based baselines in genuinely large regimes. This level of cross-scale generalization is highly relevant for practical deployment.
>
> Real industrial problems can be even more complex—flexible routing, open shop structures, sequence-dependent setups, machine groups, and additional constraints. These settings remain future work, and we plan to extend the same theory-guided pipeline (“theory → indicator → learning”) to such variants by deriving the corresponding conditions and indicator forms.
>
> ## Question 1 — Illustrations and intuition for Propositions 1–3
>
> Yes. As noted above, we will add intuitive Gantt-chart illustrations, step-by-step reasoning for each condition, and makespan changes.
>
> ## Qustions 2 and 3 — Other variants, COPs, and more complex JSSP instances
>
> Our indicators are intentionally derived for classical JSSP makespan, and the propositions do not hold once routing flexibility or additional constraints are introduced. Therefore, they do not directly transfer to flexible JSSP, open shop, or other constrained variants. Extending the method to such settings requires new theoretical conditions, which we plan to derive in follow-up work using the same “theory → indicator → learning” pipeline. This is a natural direction for applying the approach to richer industrial environments.
>
> For larger-scale evaluation, we agree that results beyond standard TA benchmarks are useful. LSI already demonstrates strong cross-scale generalization—trained on 10×10 and effective on 100×20—and we have ongoing experiments on customized 200×20 and 300×20 JSSPs with several baselines. If these runs complete during the discussion period, we will share the results immediately and include them in the appendix of the final version.

---

### Official Review · Reviewer_t1Aw · 2025-10-31

**Soundness:** 3
**Presentation:** 3
**Contribution:** 2
**Rating:** 4
**Confidence:** 5

**Summary:**

This paper introduces a new learning-driven local search technique for job shop scheduling. Its main innovation is using three theoretically-grounded, necessary conditions for improving the schedule makespan, which are converted into simple binary indicators. This approach allows a very simple MLP-based policy network to surpass more complex state-of-the-art methods that use sophisticated graph neural networks, proving the power of embedding theoretical knowledge into machine learning for combinatorial optimization.

**Strengths:**

1. The paper is well-organised and well-written in general.
2. The authors demonstrate their method's robustness via a detailed empirical evaluation performed on a range of standard JSP benchmarks.
3. The research is well-motivated and justified with empirical evidence that echoes their claims.
4. The performance is superior to existing learning-based methods.

**Weaknesses:**

1. The paper's novelty is constrained, as its core framework is heavily derived from prior works, L2S and TBGAT. It adopts their Markov Decision Process formulation and a subset of the N5 neighbourhood for the local search process. The sole apparent contribution—the application of theoretical indicators to select operation pairs—is itself adapted from existing operations research literature, further limiting the originality of the proposed method.
2. Leading neural combinatorial optimization (NCO) methods, such as L2S and TBGAT, rigorously demonstrate that their computational complexity scales linearly with problem size. Given the fundamental importance of efficiency in combinatorial optimization, the absence of a similar analysis for the proposed method diminishes the completeness of its evaluation and, consequently, its practical significance.
3. The method's limited novelty, combined with its exclusive focus on the Job Shop Problem, restricts its overall impact and broader relevance to the field.

**Questions:**

1. The paper would be strengthened by a detailed computational complexity analysis of the proposed algorithm.
2. What is the computational overhead associated with calculating the theoretical indicators in each step of the local search?
3. For large-scale problem instances (e.g., exceeding 2000 operations), could the process of computing these indicators become the primary computational bottleneck, potentially undermining the method's efficiency?
4. Given that the theoretical foundations are adapted from prior Operations Research literature, to what extent does this work contribute new theoretical discoveries, as opposed to the application of existing ones?

---

> ### Author Response · Authors · 2025-11-29
>
> We thank the reviewer for the positive comments on soundness, organization, robustness, and empirical performance, and we appreciate your patience while we prepared the detailed response. We address the concerns regarding novelty, computational complexity, and scope below.
>
> ## Weakness 1 — Limited novelty and reliance on prior work
> It is true that our MDP formulation and the overall N5-based local-search framework follow L2S and TBGAT. This was intentional: by keeping the framework identical to prior work, any performance difference can be attributed directly to our theoretical contributions rather than architectural or algorithmic confounders.
>
> The core novelty of our work lies not in redefining the MDP, but in deriving new numerical conditions specific to JSSP makespan and converting them into learnable indicator features.
> Classical OR analyses of N5 provide qualitative guidance at the critical-block level, but, to the best of our knowledge, do not offer explicit numerical necessary and sufficient conditions for makespan reduction at the level of consecutive machine operations. In contrast, we:
> 1. derive explicit EST/LST-based numerical conditions for makespan reduction for the first time,
> 2. prove an if-and-only-if result under a single critical path (Theorem 1), and
> 3. extend the analysis to multiple critical paths in the revised version (Theorems 2 and 3).
>
> These results do not appear in the OR literature in this numerical form. They are precisely what make it possible to encode theory as compact indicators and integrate them into a simple MLP policy that surpasses more complex GNN encoders.
>
> ## Weakness 2 — Missing computational complexity analysis
> We agree that the original draft did not sufficiently discuss complexity. A dedicated section will be added. In brief, our method avoids the graph-topology processing used in L2S/TBGAT—our encoder does not perform any message passing—so it incurs no graph-related overhead.
> Per iteration:
> 1. N5 generation + EST/LST update: Same as L2S/TBGAT, O(n + m).
> 2. Indicator computation:
>  If the N5 neighborhood yields k candidate pairs, each pair requires checking only the three relevant operations (the pair and its critical-path neighbors).
>  This gives O(k) complexity, and k is proportional to the number of jobs and machines → O(n+m) worst-case.
> 3. MLP policy evaluation:
>  One feed-forward pass per candidate → O(k · d).
>
> Thus, each iteration remains linear in the number of candidate pairs, matching the asymptotic behavior of GNN-based methods while relying on a far simpler encoder. Because we avoid graph-topology processing and use an MLP-based encoder, the overall runtime scales more favorably, yielding faster performance on large instances.
>
> ## Weakness 3 — Contribution restricted to JSSP
> JSSP makespan is an extremely challenging setting: classical benchmarks such as FT, ABZ, YN, SWV, and TA have been studied for decades, and even recent learning-based methods typically reduce optimality gaps only marginally. Demonstrating further improvement with a lightweight MLP-based policy is therefore non-trivial and highlights the effectiveness of our theoretical indicators.
>
> We agree that broader relevance requires going beyond JSSP. Our intention is to position this work as the first concrete instance of a theory → indicator → learning pipeline for neural combinatorial optimization, rather than as a solution limited to a single problem.
>
> We also plan to extend this pipeline to other scheduling and combinatorial optimization problems, including flexible job shop scheduling and routing problems such as CVRP. In the revised version, we will add a dedicated discussion on how the same pipeline can be instantiated for other objectives (e.g., tardiness-based criteria) and for variants such as FJSP, clarifying the generality and extensibility of the approach.
>
> ## Q1 — Complexity analysis
> Clarified in Weakness 2; a section of complexity will be added in the appendix.
>
> ## Q2 — Overhead of indicator computation
> Indicator computation is constant-time per candidate, and therefore scales linearly with the number of N5 candidates. In small instances, this overhead is of similar magnitude to a single GNN forward pass, so total runtimes become comparable. In contrast, on large instances the cost of GNN encoders grows with graph size, while the cost of our indicator evaluation does not. As a result, LSI yields noticeably better gap–time trade-offs in large-scale JSSPs. This trend is evident in Figure 5, where LSI outperforms GNN-based methods and CP on TA 50×20 and TA 100×20.
>
> ## Q3 — Theoretical novelty
> Classical OR results do not provide numerical conditions that are both (i) necessary and sufficient under explicit assumptions, and (ii) suitable for direct embedding as neural features. Theorems 1–3 in our work fill this gap, and we will emphasize this more clearly.

---

### Official Review · Reviewer_qoGT · 2025-10-31

**Soundness:** 3
**Presentation:** 3
**Contribution:** 2
**Rating:** 4
**Confidence:** 4

**Summary:**

This paper proposes LSI, a learning-based local search method for the JSS problem. The core contribution is the integration of a theoretically-derived conditions for makespan reduction directly into the policy network as binary indicators. This approach allows the use of a simple MLP-based architecture as compared to complex GNN encoders. The paper demonstrate through extensive experiments that LSI achieves better performance on standard JSS benchmarks.

**Strengths:**

The paper derives and uses necessary conditions to guide the search. The quality of the work is good with rigorous experimental evaluation against baselines. The method is scalable to larger instances by being trained only on 10x10.

**Weaknesses:**

- The primary contribution that is the hand-derived theoretical indicators based on domain knowledge.  The process of deriving such indicators for other variants of JSS or for different objectives for standard JSS (e.g., total weighted tardiness) appears non-trivial and can limit the broader applicability of this specific method without significant additional theoretical work.
- Currently the method is evaluated only on JSP. Experiments on other variants of JSP (e.g., Flexible JSP) or on other objectives would strengthen the work.

**Questions:**

- The proposed indicators are specific to JSP for makespan minimization. Could you comment on the feasibility and potential challenges of deriving a similar set of theoretical indicators for other scheduling objectives, and to other JSP variants?
- The initial solution is based on FDD/MWKR. What is the impact on performance if the initial solution is randomly initialized?
- The abstract states that LSI offers "faster inference"; however, the runtime reported in Table 1 is mostly comparable to learning-based methods like SN and IRD, and this is given that they use complex GNNs while LSI uses MILP.

---

> ### Author Response · Authors · 2025-11-29
>
> We thank the reviewer for the clear summary and for recognizing the soundness, empirical quality, and scalability of our work. We also appreciate your patience while we prepared the detailed response. We address the concerns on generality, scope, and efficiency below.
>
> ## Weakness 1 – JSSP-specific indicators and limited broader applicability
> We agree that the current indicators are derived specifically for JSSP makespan, and extending them to other variants or objectives requires additional theoretical work. Our goal, however, is not to claim that these three indicators themselves are universally reusable, but to propose a reusable “theory → indicator → learning” pipeline:
> derive necessary and sufficient conditions for when a local move improves the objective, encode them as theory-grounded indicators, and let a policy network learn to use them within local search.
>
> Classical OR analyses of N5 provide qualitative guidance at the critical-block level, but, to the best of our knowledge, do not offer explicit numerical necessary and sufficient conditions for makespan reduction at the level of consecutive machine operations. In contrast, we:
>
> 1. derive such conditions using EST/LST and critical-path structure,
> 2. prove an if-and-only-if result under a single critical path (Theorem 1), and
> 3. extend the analysis to multiple critical paths in the revised version (Theorems 2 and 3).
>
> These tailored conditions make it possible to encode the theory as binary indicators and use them effectively in a simple MLP policy. To our knowledge, this is the first work that integrates explicit scheduling-theoretic conditions directly into the policy architecture of a learning-based scheduling method.
>
> ## Weakness 2 – Evaluation only on JSP and makespan
> We acknowledge that our experiments focus on classical JSSP benchmarks and makespan. This choice is deliberate: these benchmarks remain highly challenging even after decades of study, and recent learning-based methods (L2S, TBGAT, NeuroLS, etc.) typically reduce optimality gaps only marginally. In such a regime, achieving further improvements with a much simpler architecture is non-trivial.
>
> We also plan to extend this pipeline to other scheduling and combinatorial optimization problems, including flexible job shop scheduling and routing problems such as CVRP.
>
> ## Question 1 – Feasibility for other objectives and JSP variants
> At a high level, extending our approach to other objectives or variants follows three steps:
>
> 1. Theoretical step: analyze when a local move improves the new objective (e.g., slack/due-date conditions for tardiness, load-balance or assignment conditions for FJSP).
> 2. Indicator design: encode these conditions as indicator features that remain efficient to compute over the neighborhood.
> 3. Learning step: train a policy to exploit these indicators within local search.
>
> The main challenge is theoretical rather than conceptual: deriving conditions that are both informative and computationally simple. We will add brief sketches in the discussion section to clarify how this pipeline can generalize to other JSP variants and COPs.
>
> ## Question 2 – Impact of the initial solution
> We agree that robustness to the initial solution is important. In the paper, all methods share the same FDD/MWKR initialization, following prior work, to ensure a fair comparison.
> In practice, LSI is not tied to a particular dispatcher. Because the policy is trained on a wide distribution of schedules generated during local-search trajectories, it naturally encounters diverse structures during training. As a result, the learned policy tends to reach similar gaps even when the initial schedule differs, with only convergence speed changing.
>
> We are organizing additional experiments with alternative initializations (e.g., random schedules or other dispatching rules), and will report the results in the rebuttal and in the appendix of the final version once they are consolidated.
>
> ## Question 3 – “Faster inference” vs runtime results
> We appreciate this point and agree that the wording in the abstract was too broad. In practice, runtime behaves as follows:
> On small instances, indicator computation over the N5 neighborhood can be comparable to a GNN forward pass. Thus, LSI’s runtime is similar to GNN-based methods, with the main advantage being improved solution quality.
>
> On large instances, the cost of GNN encoders grows with problem size, while our indicator evaluation remains linear in the neighborhood size with a small constant factor. In this regime, LSI achieves smaller gaps and shorter wall-clock time than GNN-based local search and even CP, making it suitable for large industrial-scale problems.
> We will therefore restrict the “faster inference” claim to large-scale settings (e.g., TA 50×20, 100×20) and clarify that on small instances LSI mainly offers better quality at similar runtime.

---

### Meta-Review · Area_Chair_enzF · 2026-01-05

**Summary:**

Reviewers' concerns are mainly around the following points:

W1. The primary contribution is the hand-derived theoretical indicators based on domain knowledge, and cannot be easily adapted to other scheduling problems besides JSP

W2. Evaluation is only on JSP

W3. The novelty is limited since the proposed method is heavily derived from prior works

W4. Lacking complexity analysis

W5. The theoretical presentation is hard to understand

W6. Improvement is marginal

**Reviewer Concerns:**

A key concern shared by all reviewers is that the theoretically derived indicator is for JSSP with makespan minimization only, and extend this procedure to other problems and objectives is not straightforward. This severe limits the scope and applicability of the proposed method. I agree with this common concern, and reviewers' responses cannot address this point.

**Reviewer Scores:**

No reviewer participated in the discussion. I found it hard for the reviewers to raise their scores.

---

### Decision · Program_Chairs · 2026-01-26

Reject